# A longitudinal characterization of sex-specific somatosensory and spatial memory deficits in HIV Tg26 heterozygous mice

Mary F. Barbe [1,2]*, Regina Loomis[2,3], Adam M. Lepkowsky[1], Steven Forman[1], Huaqing Zhao[4], Jennifer Gordon[2,3]

**1** Department of Anatomy and Cell Biology, Lewis Katz School of Medicine, Temple University, Philadelphia, PA, United States of America, **2** Comprehensive NeuroAIDS Center, Lewis Katz School of Medicine, Temple University, Philadelphia, PA, United States of America, **3** Department of Neuroscience and Center for Neurovirology, Lewis Katz School of Medicine, Temple University, Philadelphia, PA, United States of America, **4** Department of Clinical Sciences, Lewis Katz School of Medicine, Temple University, Philadelphia, PA, United States of America

\* mary.barbe@temple.edu, mbarbe@temple.edu

**Data Availability Statement:** All relevant data is within the manuscript and its Supporting Information.

## Abstract

The pathogenesis of human immunodeficiency virus associated neurological disorders is still not well understood, yet is known to result in neurological declines despite combination anti-retroviral therapy. HIV-1 transgenic (Tg26) mice contain integrated non-infectious HIV-1 proviral DNA. We sought to assess the integrity of neurocognitive function and sensory systems in HIV-1 Tg26 mice using a longitudinal design, in both sexes, to examine both age- and sex-related disease progression. General neurological reflexive testing showed only acclimation to repeated testing by all groups. Yet, at 2.5 months of age, female Tg26 +/- mice showed hyposensitivity to noxious hot temperatures, compared to wild types (both sexes) and male Tg26 +/- mice, that worsened by 10 months of age. Female Tg26 +/- mice had short-term spatial memory losses in novel object location memory testing at 2.5 and 7 months, compared to female wild types; changes not observed in male counterparts. Female Tg26 +/- mice showed mild learning deficits and short- and long-term spatial memory deficits in olfactory and visually cued Barnes Maze testing at 3 months of age, yet greater learning and memory deficits by 8 months. In contrast, male Tg26 +/- mice displayed no learning deficits and fewer spatial memory deficits (mainly heading errors in nontarget holes). Thus, greater sex-specific temperature hyposensitivity and spatial memory declines were observed in female HIV Tg26 +/- mice, than in male Tg26 +/- mice, or their wild type littermates, that increased with aging. Additionally, tibial bones were examined using ex vivo micro-CT after tissue collection at 11 months. Sex-dependent increases in bone volume and trabecular number were seen in males, matching their greater weights at this age. These results indicate that HIV-1 Tg26 mice is a promising model in which to study neuropathic mechanisms underlying peripheral pathology as well as cognitive deficits seen with HIV.

**Funding:** Acknowledgements. Research reported in this publication was supported by the National Institute of Mental Health of the National Institutes of Health under Award Number P30 MH092177 to KK (https://www.nimh.nih.gov/index.shtml). The content is solely the responsibility of the authors and does not necessarily represent the official views of the National Institutes of Health.

**Competing interests:** The authors have declared that no competing interests exist.

## Introduction

The pathogenesis of human immunodeficiency virus (HIV) associated neurological disorders (HAND) is still not well understood, yet is known to result in a decline in everyday functioning despite the introduction of combination anti-retroviral therapy (cART). Neurocognitive and neurodegenerative changes due to the impact of HIV-1 infection have been modeled in rodents in several different manners. Utilizing infectious virus to study HIV has several limitations due to the species specificity of HIV-1, which replicates only in cells of human origin. Infectious studies in rodents are therefore limited and require engraftment of human cells to serve as host for HIV (i.e., humanized mice), the use of an analogous retrovirus native to the host (i.e., murine retrovirus), or use of pseudotyped virus, such as murine EcoHIV, which allows the initial stages of infection of mouse cells but no further rounds of viral replication.

Examining the pathogenesis of HIV-1 neurological disease in small animals has been approached using transgenic rodents. Transgenic rodents, with stable copy number and integration sites provide a suitable background on which to test the effects of the transgene across the lifespan of the animals. Toward this end, HIV transgenic mice and rats have been generated which contain the identical transgene encoding the entire NL4-3 HIV-1 genome with a deletion of a 3 kb region of the gag/pol genes (NL4-3Δgag/pol). Thus, the transgene encodes the env gene as well as the genes for the regulatory proteins tat, rev, vif, vpr, and vpu under the control of the native viral 5' and 3' LTRs. The resulting models, known as the HIV transgenic (Tg) rat and the HIV Tg26 mouse, have been utilized by many groups to model several pathologies described in HIV-1 infected individuals [1, 2]. Previous studies have detected expression of viral transcripts in these animals across their lifespan. This is thought to mimic the low level production of HIV proteins believed to be ongoing in HIV-1 infected individuals well controlled on anti-retroviral therapy and therefore supports the use of these models for examining the neurological impact of chronic long term infection with HIV-1.

While behavioral studies in HIV Tg26 mouse have just begun [3], neurological deficits of HIV transgenic rats have been intensively studied by several groups [4–11]. In the initial mouse study, a cross-sectional design was used to examine 8–10 month old (i.e., mature) mice in spatially cued Barnes maze testing [3]. Spatial learning deficits during acquisition trials were observed in that study of mature female Tg26 +/- mice, but not in mature male Tg26 +/- mice, relative to same sex wild type mice. Also, short-term spatial memory retention deficits were observed in female Tg26 +/- mice, while both short term and long-term spatial memory deficits in male Tg26 +/- mice, relative to same sex wild type mice [3]. In vivo and in vitro studies on these Tg26 +/- mice show early and late-stage neurogenesis deficits, such as deficiencies in dendritic arborization, length and spine density on neurons in hippocampal dentate neurons [12]. In the rat HIV Tg26 model, several progressive behavioral deficiencies have been noted, including deficits in locomotor activity, disruption in working memory, sensory gating and attention, as well as other neurocognitive changes [13–15]. These deficiencies are believed to be driven by HIV proteins expressed at low levels that then induce neurotoxicity via underlying mechanisms, including neuroinflammation, alterations in dopamine and mu opioid receptor levels, and increased cytokine and chemokine levels during development [15–17].

The HIV Tg26 transgenic mouse model was originally generated on an FVB/N background by microinjection of a transgene encoding the entire NL4-3 HIV-1 genome with a deletion of a 3 kb region of the gag/pol genes (NL4-3Δgag/pol) [1]. Tg26 mice contain the HIV-1 genome under the control of the native HIV-1 5' and 3' LTR regulatory sequences stably integrated into the mouse genome [1, 2, 18, 19]. Characterization of copy number and insertion site have revealed that the line contains 10 tandem head-to-tail copies of the transgene NL4-3Δgag/pol stably integrated within the mouse genome at a site mapped to locus C2 of chromosome 8 by

in situ hybridization [18]. Expression of viral transcripts and low level expression of viral proteins have been reported in the kidney, spleen, thymus, and lymph nodes prior to disease onset [1, 20, 21]. Western blot analysis has shown the viral protein, gp120, in spleens and lymphomas at low levels in healthy animals and at increasingly higher levels in diseased tissues [20, 22]. Nef expression has been observed in the lymphoma tissues and rev was observed in sclerotic glomeruli in the kidney. As a consequence, mice heterozygous for the transgene develop many clinical features of HIV infection and AIDS including a disease similar to HIV-associated nephropathy (HIVAN), as well as changes in other tissues and organs, including, inflammation, B cell lymphomas, skin lesions consistent with cutaneous papillomas, cardiomyopathy, and muscle wasting between 3 and 6 months of age on the FVB/N background [19, 20, 23, 24]. Interestingly, degenerative changes in bone, another key organ, have been observed in mature HIV-1 Tg heterozygous rats (NL4-3Δgag/pol Fisher344 rats), compared to mature wild type rats However, possible bone degeneration that may occur as a result of increased viral load, altered general physical characteristics or neurological issues has yet to be examined in HIV mouse models.

Thus, HIV-1 transgenic (Tg26) mice were created using non-infectious HIV-1 proviral DNA, and may serve as a model for investigating the impact of HIV-1 on neurodegeneration. Due to the early lethality of the mice, we crossed them onto the C57BL/6J background where the disease is more stable and mice routinely survive up to 12 months, though we still observe signs of skin lesions and some of the other pathologies of the original line.

With these mice, we sought to now assess, for the first time, the integrity of neurocognitive function and sensory systems in HIV Tg26 +/- mice using a longitudinal design from 1 through 10 months of age, in both males and females, in order to examine both age- and sex-related disease progression. Additionally, we collected tibial bones from the HIV Tg26 +/- mice and their wild type mouse littermates at terminal endpoints and to examine for the first time if HIV Tg26 +/- mice have degenerative bone changes, using microCT methodology.

## Materials and methods

### Animals

The Temple University Institutional Animal Care and Use Committee approved all experiments, in accordance with the guidelines laid down by the National Institute of Health. We generated Tg26 (+/-) mice on a complete C57BL/6J background by backcrossing FVB/N-Tg (HIV)26Aln/PkltJ mice (Jackson Lab, #022354) with C57BL/6J mice (Jackson Lab, #000664) at a minimum of 6 generations of back crossing, as described previously [3]. This same backcrossing procedure was performed in a similar manner as with HIV-1 transgenic rats to prolong their life expectancies [25]. Throughout the study, the Tg26 mice were maintained as heterozygotes (+/-), since Tg26 homozygous (+/+) mice rarely survive to weaning [26].

Forty-one mice were used from a total of seven litters, and were divided into 4 groups based on sex and genotype at 1 and 2 months of age: 10 female wild type -/- mice, 8 female Tg26 +/- mice, 10 male wild type -/- and 10 male Tg26 +/- mice. At 7 months of age, 10 female wild type -/- mice, 8 female Tg26 +/- mice, 8 male wild type -/- and 5 male Tg26 +/- mice were included (seven male mice were removed from the study for other investigations). Numbers of animals per group and per test are indicated hereafter in figures or their legends. After weaning at day 21, mouse pups were handled for one week before onset of experiments. Mice were group-housed with same-sexed siblings on a 12:12 light dark cycle with food and water ad libitum. After being moved to the animal behavioral testing facility, mice were allowed at least one hour to acclimate before behavioral testing. Experimenters were naive to animal genotype

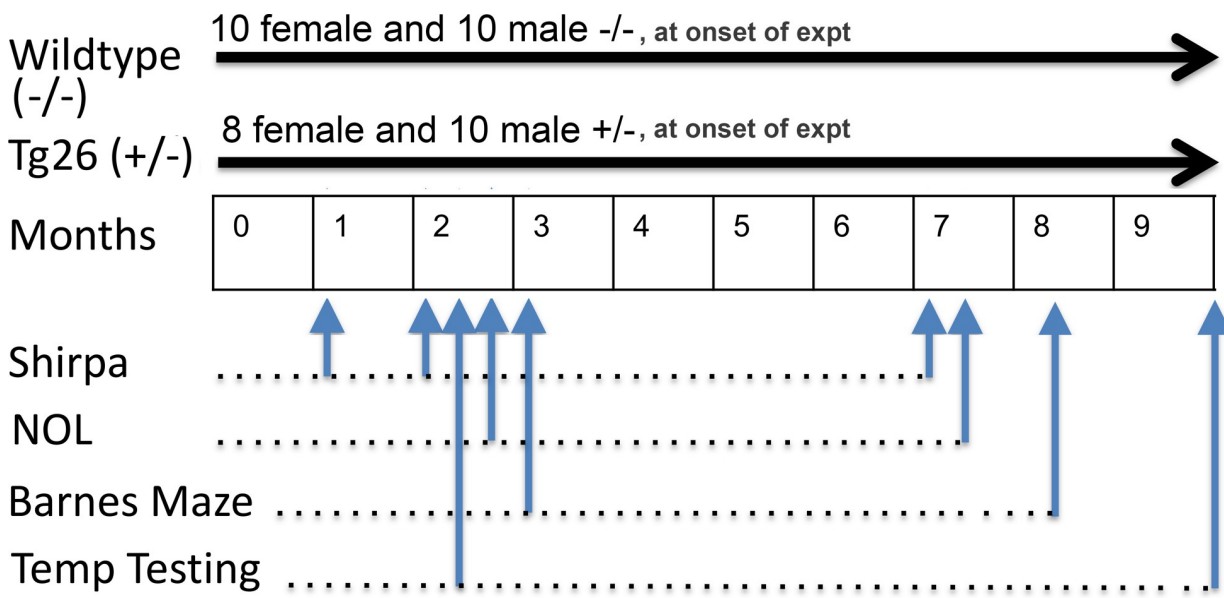

**Fig 1. Longitudinal experimental design.** Mice were tested at several points across time s shown for general physical characteristics and neural reflexes (SHIRPA), novel object location (NOL), Barnes Maze and Temperature Place Preference testing. Numbers per group were as follows: At 1 and 2 months of age, there were 10 female wild type -/- mice, 8 female Tg26 +/- mice, 10 male wild type -/- and 10 male Tg26 +/- mice. At 7 months of age, 10 female wild type -/- mice, 8 female Tg26 +/- mice, 8 male wild type -/- and 5 male Tg26 +/- mice were included (seven male mice were removed from the study for other investigations).

during all analyses. Animals underwent repeated behavioral testing for various neurologic, sensory or memory assays across time (1 through 10 months of age; Fig 1).

### General health and neurological testing

General health screenings, neurological assays of reflexive motor and sensory circuits, and reactions to handling, were performed on the mice at 1, 2 and 7 months of age (See Fig 1 for Design) using modified primary SHIRPA screening methods [27–33] that was modified from the original primary SHIRPA screening developed by Irwin in 1968 [34]. See Table 1 for details of the specific tests performed and their scoring system. For consistency, this is the same primary screening method utilized in our prior study on Tg26 +/- mice [3] and a related mouse model [35]. These assays were performed in an isolated and quiet room.

Briefly, first, the mice were placed into a clean cage for observations of gross neurological functions [32], (for 3 minutes per testing session). As shown in Table 1, transfer arousal behaviors and locomotor/spontaneous activity were scored during this observational period. Bizarre/abnormal behaviors, such as wild running, freezing, sniffing, licking, rearing, excessive grooming, circling, tremor, excessive head bobbing, stays in perimeter, and climbing were recorded on occurrence (the number of such individual behaviors per session was summed per test period), as was any occurrence of tremors. Body position, gait, pelvic elevation and tail elevation were scored. Incidence of vocalization, biting, aggression, defecation (number of fecal boli) and urination during this 3 minute test period was also recorded.

Next, mice underwent a general physical examination for indices of general health (Table 1). All animals were weighed and measured at 1, 2, 3 and 7 months of age. Skin color and texture, coat quality, piloerection, whisker condition, palpebral closure, lacrimal and nasal discharge, or respiration was recorded and scored. Presence of any wounds, masses or morphological abnormalities of the head, body, limb and tail were recorded. Body, limb and abdominal tone were assayed using light restraint at this time. Next, several sensory and motor

**Table 1. Modified primary SHIRPA screen: Tests and scoring method.**

| Observation in a clean cage arena (3 minutes) | |
|---|---|
| Transfer arousal (7 point scale) | Coma (0); Immediate movement (1), momentary freeze (2); Brief freeze (3) extended freeze than moderate movement (4); Prolonged freeze (> 10 sec) then slight movement (5); Extremely excited/manic (6) |
| Locomotor/spontaneous activity (6 point scale) | None (0); Vigorous scratch, groom, moderate movement (1); Casual scratch, groom, slow movement (2); Vigorous rapid/darting movement (3); Extremely vigorous movement (4); Jumping excessively (5) |
| Bizarre/abnormal behavior (scored as sum of behaviors observed) | Wild running, freezing, sniffing, licking, rearing, excessive grooming, circling, tremor, excessive head bobbing, stays in perimeter, climbing, other (describe) |
| Body Position | Rounded (normal, 0); hunched (1) |
| Respiration rate (5 point scale) | Normal (0), labored, retching, dyspneic, gasping |
| Tremor (3 point scale) | Absent (0), mild (1), marked (2) |
| Gait (4 point scale) | Normal (0), fluid but abnormal (1), limited movement only (2), incapacity (3) |
| Pelvic Elevation (4 point scale) | Normal (~ 3 mm, 0), barely touches ground (1), markedly flatted (2), elevated (> 3 mm, 3) |
| Tail Elevation (3 point scale | Horizontal (norm, 0), dragging (1), elevated (2) |
| Defecation | Number of fecal boli |
| Urination (relative amount; 3 point scale) | Small, medium, large amount |
| General Health/Morphology, using light restraint as needed | |
| Body weight (g) | |
| Body length (nose tip to tail tip; mm) | |
| Skin color and texture of pinna, food pads, tail | Normal (0), abnormal (1); describe if abnormal |
| Fur color/pattern | Description |
| Fur condition (4 point scale, 0–4) | Normal (0), matted, discolored, hair loss |
| Evidence of grooming (2 point scale) | Present (0), absent (1) |
| Piloerection, undisturbed (2 point scale) | Absent (0), present (1) |
| Whisker condition (4 point scale, 0–4) | Normal (0), sparse, short or missing |
| Palpebral closure (4 point scale, 0–4) | Normal (0), swollen, red, or crusty/weeping |
| Lacrimation | Absent (0), present (1) |
| Nasal discharge | Absent (0), present (1) |
| Wounds | Absent (0), present (1); location if present |
| Morphology of head (ears, snout, mouth and teeth), body/trunk, tail, forelimbs, hindlimbs, forepaws, hindpaws, genitals | Normal (0), abnormal (1); describe and location if abnormal |
| Abnormal masses or growths | Absent (0), present (1); location if present |
| Body tone (3 point scale) | Flaccid/low (0), normal (1), high (2) |
| Limb tone (3 point scale) | Flaccid/low (0), normal (1), high (2) |
| Abdomen tone (3 point scale) | Flaccid/low (0), normal (1), high (2) |
| Above the Arena | |
| Positional passivity (Struggle response) | Does not struggle (0); Struggles when restrained: by tail (normal, 1), by loose scruff at neck (2), held supine (3), by hindlimbs (4) |
| Trunk curl response (normal is attempting to "crawl up" its own body when lifted by tail) | Present (0), absent (1) |
| Hindimb extension reflex during suspension | Present (0), absent (1) |
| Visual placing response/visual forepaw reach when mouse held 25 mm above clean cage surface and then lowered (4 point scale) | Before vibrissae contact (0), upon vibrissae contact (1), upon nose contact (2), none (3), early vigorous extension (4) |

*(Continued)*

**Table 1.** (Continued)

| **Sensory (sensorimotor) Reflexes** | |
|---|---|
| Visual placing response | See sensory reflexes above |
| Corneal response (eye blink) to light touch with cotton swab | Blink/heat twitch (0), no response (1) |
| Pupillary response to light, and presence of cataracts | Present (0), absent (1) |
| Pinna (ear) reflex to light touch with cotton swab | Ear/head twitch (0), no response (1) |
| Whisker twitch response (3 point scale) to light touch with cotton swab | Whiskers stop moving and animal orients to touch (0), whiskers stop moving (1), no response (2) |
| Hindpaw toe pinch (4 point scale) | Brisk/rapid withdrawal (0), slight withdrawal (1), moderate withdrawal (2), very brisk with repeated extension & flexion suggestive of pain (3), none (4) |
| Touch escape (5 point scale)–modified since mice began to crawl into testers hands during our longitudinal testing paradigm. | Mild (0), moderate (1); vigorous (2); no response (-1), crawled into tester's hand (-2) |
| Acoustic startle response to a clicker | Startle Preyer reflex (0), no response (1) suggestive of hearing loss |
| **Motor Abilities** | |
| Gait in arena | See first 5 minute arena tests |
| Limb or paw paralysis/lameness | Absent (0), present (1); description |
| Righting reflex | Normal (~ 1 sec, 0), No response (0) |
| Negative geotaxis | Normal (turn and locomote up the slope (0); No turn, slip or no locomotion (1) |
| Open field speed | Quantified using Any Maze software |
| Open field total distance | Quantified using Any Maze software |
| **Additional Comments** | |
| Vocalization during testing | Absent (0), present (1) |
| Biting during testing/irritability | Absent (0), present (1) |
| Aggression during testing | Absent (0), present (1) |
| Defecation during testing | Number of fecal boli |
| Urination during testing (relative amount; 3 point scale) | Small, medium, large amount |

reflexes were measured [28, 36] (Table 1). With the mouse elevated above the arena, positional passivity was scored with increasing intensity of restraints. Trunk curl, hind-limb extension and visual placing reflexes were assessed by gently restraining the animal by the tail, elevating them above the arena (~25 mm) and then lowering the mouse to the testing surface rapidly. Corneal blink, pupillary responses and potential presence of cataracts, pinna reflex, whisker response were tested in awake and freely moving mice by gently brushing eyelids, ears, or whiskers with a sterile cotton swab. Acoustic startle in response to a training clicker (<70 dB) was scored [35]. Escape to touch was performed by approaching the mouse with a bent index finger. The righting and toe pinch reflex assays were performed last. Since the animals began responding with clear habituation to the tester across time, the escape to touch scoring system had to be modified to: Mild (0), moderate (1), vigorous (2), no response (-1), and crawled into tester's hand (-2) (Table 1). Vocalization, biting, aggression, defecation and urination during the above testing was also noted on occurrence and scored as described in Table 1. The equipment was cleaned between mice (Saniwipes).

## Temperature place preference testing

Temperature sensitivity was tested using a temperature place preference assay at 2.5 and 10 months of age (Fig 1). Temperature place preference to a room temperature plate (25˚C)

versus a plate that was increasing from warm to hot temperatures (25–41˚C) was tested across a 35 minute timespan (with 5 minutes (300 sec) per temperature), as previously described [37]. At least 24 hours later, temperature place preference to a room temperature plate versus a plate that was decreasing from cool to cold temperatures (20–12˚C) was similarly tested.

## Learning and memory tests

**Novel object location memory.** Novel Object Location memory was tested at 2.5 months and 7 months of age (See Fig 1 for Design). For this, a spatial memory task was used in which mice were first allowed to explore three objects in an open field box, and then retested at one hour later and then again 1 week later, to determine their ability to recognize that one object had been moved (thus, short-term and long-term spatial memory test, respectively), as previously depicted [35]. In detail, the objects used were equal in shape and size (65 cm high x 60 cm long x 30 cm deep) and were plastic objects with no natural significance to mice. Dark curtains were hung to form a 177 cm x 177 cm square around the maze to exclude extraneous cues. The mice were first habituated to an empty open field arena for 5 minutes. Twenty-four hours later, mice were permitted to explore three objects placed in a diagonal arrangement in three 5-minute acquisition trials, with 3 minutes between each trial. Between each acquisition trial, mice were removed and the arena and objects cleaned. One hour later, memory retention was probed by returning the mouse to the arena in which one object had been moved to a new location, and given 3 minutes to explore the objects. Nose-point tracking was used to define exploration of an object [38]. Memory retention was probed again 1 week later (this time point was chosen to match the timing of the Barne's maze testing). Discrimination between the objects was calculated using a discrimination ratio, calculated as the absolute different in the time spent exploring the novel and familiar objects, divided by the total time spent exploring the objects [39–41]. This calculation takes into account the individual differences in the total amount of exploration [39–41].

**Spatial and olfactory cued Barnes maze testing.** Thereafter, short-and long-term spatial memory abilities were tested using the more robust Barnes maze testing method at 3 and 8 months of age (See Fig 1 for Design) in which both spatial and olfactory cues were provided. In detail, a commercially available Barnes maze was used (Stoelting Co.). These experiments were conducted between 1 hour after and 1 hour before daily light-dark transitions during the light cycle. Dark curtains were hung to form a 177 cm x 177 cm square around the maze to exclude extraneous cues. One of four different spatial cues, consisting of simple geometric figures (plus, square, circle, triangle) was hung on each of the four curtains. Since there was a concern that the Tg26 mice might develop opaque cataracts, as seen in some transgenic mice and HIV-1 Tg rats [17, 20, 42], we modified the Barnes maze test to also include four distinct scents (peppermint, vanilla, hot sauce and almond) placed on gauze squares in small plastic perforated boxes (paraffin embedding cassettes #B1000729AQ, ThermoFisher) in one of each of four holes (target hole, opposite hole (R/L 10), east central hole (R5) and west central hole (L5)) during the acquisition trials. Between each of these trials, scent cues were removed and the tabletop and holes cleaned with 70% isopropanol.

Mice were transported to the testing room and allowed 1–2 hours each testing time point to acclimate to the new room. The animal was placed into the center of the table using a glass beaker, which served as the starting cylinder, and allowed to acclimate for 5–10 seconds. A buzzer, floodlight and fan were turned on as aversive stimuli (particularly needed in these longitudinal studies in which animals acclimated to the tester as a result of the repeated testing) and the glass cylinder was lifted, releasing the mouse. Camera recording began simultaneously. The habituation trial then began and lasted for 1 minute, before the experimenter used the starting

cylinder to gently guide the mouse to the target hole. Upon entry, the escape tunnel was then covered, the aversive stimuli were terminated and the recording halted, and the mouse rested for one minute before moving the mouse to a holding cage for 3 minutes during which time the maze was cleaned and scented gauze pads replaced with new ones.

Acquisition training began one hour later, with mice being placed on the maze and permitted to explore for 5 minutes. If the mouse did not escape the maze into the target hole (defined as all four paws leaving the surface of the platform) within the 5 minute trial, the experimenter guided the mouse to the target escape hole as before and the mouse allowed to rest for 1 minute before being returned to its holding cage. Four acquisition trials were performed per day, with 3 minutes between trials, for 4 days, for a total of 16 acquisition trials across the four days. The table was cleaned and scented gauze pads replaced with fresh ones between trials.

The first memory retention probe trial occurred at 24 hours later and again 7 days after the last training trial, i.e., on days 5 and 12 of Barnes maze testing. For this, the escape tunnel was removed from the target hole and replaced with a shallow tray to close off the escape tunnel so that it mimicked the other closed holes. Olfactory cues were also not provided during probe testing. The ability of the mice in finding the target hole was tested in a 5 minute trial with all adverse stimuli ongoing until trial end. All trials were recorded using a CCD monochrome camera and analyzed using ANY-Maze tracking software. Using nose-point tracking, the number of head entries into each hole, head time in the target hole, duration of each visit to the target hole, distance to $1^{st}$ entrance to the target hole, path efficiency to the target hole, number of head entries into the target quadrant, head time in the quadrant target, latency to head entry into the target quadrant, and average speed around the maze and distance traveled were quantified from the recording using AnyMaze software.

Specifically, path efficiency is a measure performed by the AnyMaze software and is an index of the efficiency of the path taken by the animal to get from the first position in the test (i.e., the center site of the maze where they are placed initially) to the last position (i.e., the target). For this, the software divides the straight line distance between the first position in the test and the last position by the total distance travelled by the animal during the test. A value of 1 indicates perfect efficiency—the animal moved in a straight line indicative of a "direct search"—values less than 1 indicate decreasing efficiency. The lowest value would be indicative of the most inefficient "serial search". The proportion of animals per group exhibiting a direct, serial versus random strategy for reaching the target hole was also manually scored from the AnyMaze plot tracts.

## Ex vivo microCT of tibial bones at 11 months of age

At 11 months of age, animals were euthanized using $CO_2$ followed by cervical dislocation. Tibial bones were then collected, cleaned of soft tissues, and fixed in 10% formalin for 4 days before storage in phosphate buffered saline (PBS) with sodium azide until undergoing ex vivo microCT analysis (n = 5-8/group) using a Skyscan 1172, 12 megapixel, high-resolution cone-beam microCT scanner (Bruker, Kontich, Belgium), using the following settings: a pixel resolution size of 3.88 μm, x-ray source spot size of 300 nm, Al 0.5mm filter, voltage of 58 kV, current of 100 μA, rotation step of 0.50°, and a frame averaging of 6. Each scan was approximately 35 min per tibial bone. During reconstruction of the images using cone-beam reconstruction software based on the Feldkamp algorithm (Bruker Skyscan NRecon), a ring artifact correction of 10, and a beam hardening correction of 40% were applied to all samples. This process yielded more than 1336 tomographic sections, 3.88 μm in thickness, in the axial plane, for each tibial bone. MicroCT analysis of the proximal tibial bone was performed, as previously described for mouse tibial bones [43], and briefly below. Using CTAn software (Bruker

Skyscan), trabecular bone was separated from cortical bone with a region of interest tool. Trabecular morphometric traits were computed from binarized images using direct 3D techniques that do not rely on prior assumptions from the underlying structures. The volume of interest for trabecular microarchitectural variables was bounded to the endocortical margin, starting 1.5 mm from the proximal tibial condyles in the direction of the metaphysis, and then extending from this position for 250 slices (1.5 mm). Upper and lower thresholds of 255 and 80 were used to delineate each pixel as "bone" or "nonbone," and trabecular bone volume per total volume (BV/TV), mean trabecular thickness (Tb.Th.), mean trabecular number (Tb.N.), and mean trabecular separation (Tb.Sp.) indices were computed. Trabecular tissue mineral density and cortical bone mineral density of the proximal tibias were determined using Bruker-Skycan and CtAn recommended methods.

## Statistics

Graph-Pad Prism software (version 8 for Mac OS, San Diego, CA) was used for all statistical analyses, using an α level of 0.05. Statistics were further confirmed by SAS 0.4 (SAS Institute Inc., Cary, NC). Three-way mixed-effects models (REML models) with repeated measures (and Geisser-Greenhouse corrections) were used to analyse weight and all general neurological testing outcomes (using the factors: age, sex and genotype), temperature differences (factors: sex, genotype and temperature), as well as novel object location spatial memory differences (factors: sex, genotype and retention time) between the four groups (female wild type -/-, female Tg26 +/, male wild type -/-, and male Tg26 +/-). For further comparison of novel object location memory test, mixed-effects models with repeated measures (and a Geisser-Greenhouse correction) were performed after separating data by sex or genotype. Barnes maze statistical analyses were separated by the acquisition phase (to analyze learning ability) versus retention probe phases (to analyze short and long-term spatial memory, as previously described [3]) prior to similar three-way mixed-effects model analyses (factors: sex, genotype and acquisition trial or retention probe time). Tukey's multiple comparison tests were used for post hoc comparisons, with adjusted p values reported. Spearman's correlations were used to correlate body weight with tibial bone attributes. Results were expressed as mean ± SEM.

## Results

### General sensorimotor reflex and neurological testing showed only acclimation to repeated testing across time

We first examined general transfer, sensorimotor and neurologic reflexes in mice of both genotypes and sexes across time in order to inform interpretation of further assays (Fig 2). There were no differences in toe pinch or eye blink responsiveness across time or between groups (Fig 2A and 2B). A visual placing test showed reduced responses in general with repeated testing across time (age: $F_{(1,75)} = 8.81$, p = 0.03). Post hoc analysis showed that female and male wild type (-/-) mice had the greatest reductions in visual placing scores at 7 months of age, compared to their responses at 1 month (p = 0.006; Fig 2C), likely due to acclimation to the test since the eye blink response was normal, and the results of the next assay. Escape to touch reflexes also showed reduced responses with repeated testing across time (age: $F_{(1,44)} = 46.06$, p<0.0001). Post hoc analysis showed that each group of mice had lower scores at 7 months, compared to their responses at 1 and 2 months (p<0.01 each; Fig 2D). By 7 months, mice of each sex and genotype had acclimated to the tester by 7 months of age and crawled into the tester's hands rather than escaping, resulting in negative "escape to touch" scores. Differences in transfer arousal behaviors (in the first 15 seconds after transfer to the test chamber)

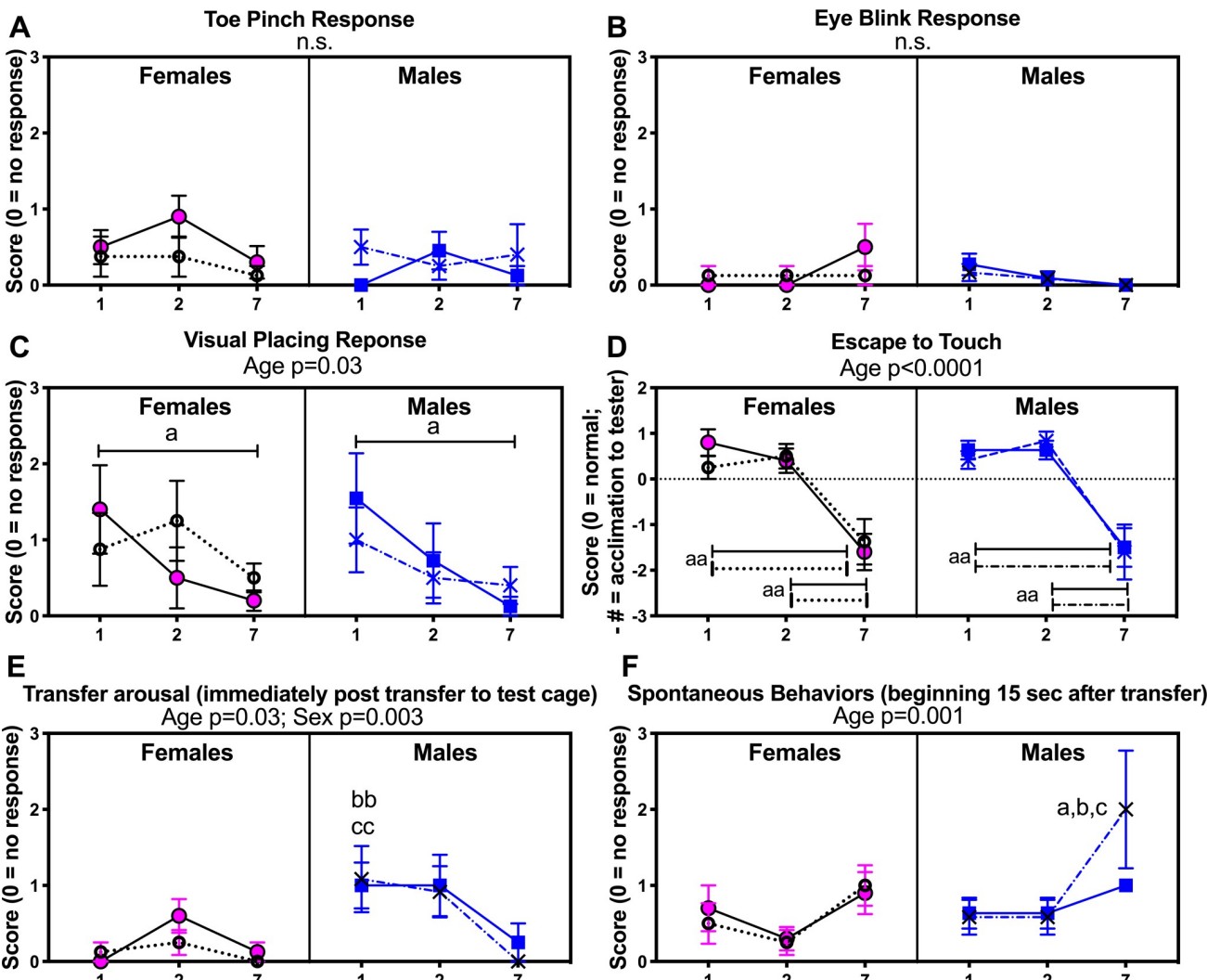

**Fig 2. Neurological reflexive testing.** Select neural reflexive results from the general neurological health screenings, in wild-type mice -/- and heterozygous Tg26 +/- mice of both sexes, across time (1, 2 and 7 months of age). (A) Responses to toe pinch. (B) Eye blink response (tester approaches mouse's eyes with a sterile cotton swab). (C) Responses to a visual placing test. (D) Escape to touch (tester approaches mouse with tester's index finger). (E) Transfer arousal immediately after transfer to the test cage, with higher number indicating higher arousal behaviors. (F) Spontaneous behaviors (beginning 15 sec after transfer), with high number indicating more vigorous/excitable behaviors. Mean ± SEM shown. Not significant (n.s.); a and aa: p<0.05 and p<0.01, compared to values obtained for these same animals at 1 mo; b: p<0.05 and bb: p<0.01, compared to opposite sex of same genotype; c:p<0.05 and cc: p<0.01, compared to opposite sex of opposite genotype. Numbers/group: At 1 and 2 months of age, there were 10 female wild type -/- mice, 8 female Tg26 +/- mice, 10 male wild type -/- and 10 male Tg26 +/- mice. At 7 months of age, there were 10 female wild type -/- mice, 8 female Tg26 +/- mice, 8 male wild type -/- and 5 male Tg26 +/- mice.

were observed (age: $F_{(2, 82)}$ = 3.86, p = 0.03; and sex: $F_{(1, 99)}$ = 9.06, p = 0.003), with males of both genotypes showing higher transfer arousal behaviors, compared to females (p = 0.01 each) at 1 month (Fig 2E). Lastly, differences in spontaneous behaviors (beginning 15 sec after transfer) were observed (age: $F_{(2, 95)}$ = 9.05, p = 0.0003; and sex: $F_{(1, 101)}$ = 3.89, p = 0.051), with Tg26 +/- males showing higher spontaneous activity at 7 months, compared to the other groups (p<0.05 each, Fig 2F).

Remaining sensorimotor and neurological tests showed no genotype or sex differences at any time point (S1 and S2 Figs). No cataracts were observed either at birth or over time, in any group. However, male Tg26 +/- showed increased incidence of patchy and gray hair at 7

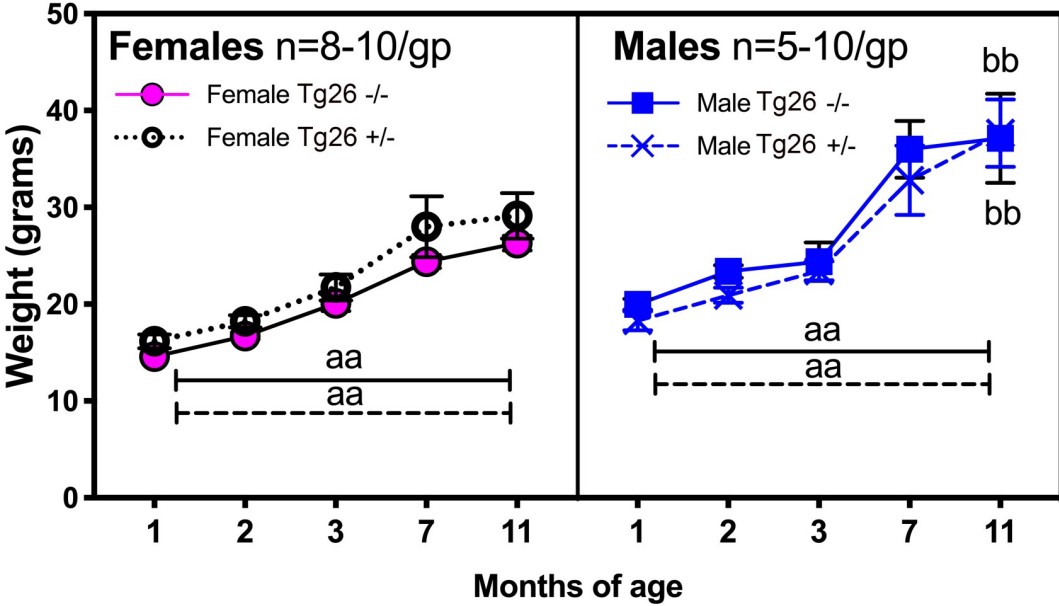

**Fig 3. Body weight differences by sex and across time.** aa: p<0.01, compared to values obtained for these same animals at 1 mo (same genotype); bb: p<0.01, compared to opposite sex yet same genotype. Numbers/group: At 1 and 2 months of age, there were 10 female wild type -/- mice, 8 female Tg26 +/- mice, 10 male wild type -/- and 10 male Tg26 +/- mice. At 7 months of age, there were 10 female wild type -/- mice, 8 female Tg26 +/- mice, 8 male wild type -/- and 5 male Tg26 +/- mice.

months of age than same aged male wild type mice. All mice consistently gained weight from 1–7 months of age ($F_{(4,118)}$ = 79.51, p<0.0001), although male mice gained more weight than female mice, regardless of genotype (sex: $F_{(1,25)}$ = 27.71, p<0.0001; age x sex interaction: $F_{(4,118)}$ = 3.74, p<0.0001; Fig 3). There were no significant weight differences between the genotypes (p = 0.86).

### Female Tg26 +/- mice show early and continued noxious heat hyposensitivity, male Tg26 +/- mice show later

Temperature preference/avoidance behaviors were tested at 2.5 and 10 months of age (Fig 4). At 2.5 months of age, a significant temperature place preference x sex interaction effect was observed ($F_{(4,132)}$ = 2.73, p = 0.03; Fig 4A). Posthoc comparisons showed that the 2.5 mo old female Tg26 +/- mice spent significantly more time on the variable plate when set at 41˚C, compared to the other groups (Fig 4A), indicative of hyposensitivity to this temperature by young adult female Tg26 +/- mice. By 10 months of age, a significant temperature preference x sex interaction effect was observed ($F_{(4,125)}$ = 3.33, p = 0.01) as was a genotype difference ($F_{(1,125)}$ = 6.401, p = 0.01; Fig 4B). Posthoc comparisons showed that mature female Tg26 +/- mice spent significantly more time on the variable plate set at 41˚C, compared to same-aged female wild type -/- and male Tg26 +/- littermates (p<0.05 each; Fig 4B), as well as more time on the plate when at 45˚C, compared to male wild type -/- littermates (p<0.01; Fig 4B), indicating hyposensitivity to these temperatures by female Tg26 +/- mice. Interestingly, the male Tg26 +/- mice also showed more time on the 45˚C plate, compared to male wild type -/- mice

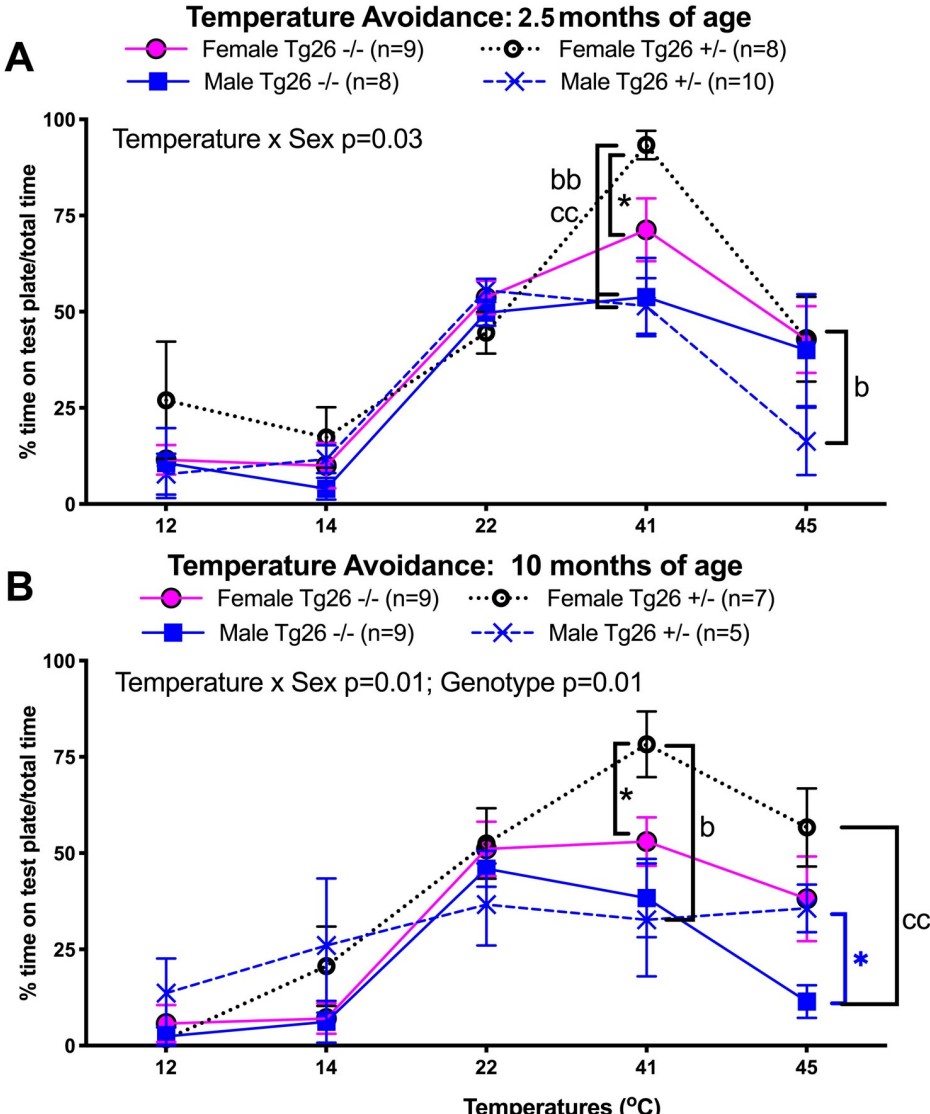

**Fig 4. Temperature sensitivity/aversion, assayed using a two-plate temperature place preference testing instrument at 2.5 and 10 months of age, in wild-type mice -/- and heterozygous Tg26 +/- mice of both sexes.** (A) Results when tested at 2.5 months of age. (B) Results when tested at 10 months of age. Mean ± SEM shown. *: p<0.05, compared to wild type -/- mice of same sex; b and bb: p<0.05 and p<0.01, compared to opposite sex of same genotype (opposite sex); cc: p<0.01, compared to opposite sex of opposite genotype (opposite sex and genotype). Number of animals per sex, genotype and age, are indicated in the keys of panels A and B.

(p<0.05), although similar to female wild type littermates. No age-related differences were observed. In summary, there were early temperature sensory nerve or processing deficits in female Tg26 +/- mice that were further enhanced by 10 months of age. Hence, female HIV-1 Tg26 +/- mice displayed progressive hot temperature hyposensitivity (hyposensitivity to 41˚C at 2.5 months of age that progressed to also include hyposensitivity to 45˚C by 10 months of age), compared to wild type littermates.

## Tg26 +/- female mice show short-term spatial memory deficits

We began neurocognitive testing with a simple spatial location assay that required only a short training period, the novel object location memory assay (Fig 5). When both genotypes, sexes,

and time (age at testing and memory retention probe times) were combined into a mixed-effect model, only trends towards genotype ($F_{(1,26)}$ = 3.91, p = 0.07; Fig 5A) or sex x genotype interaction ($F_{(3,74)}$ = 2.24, p = 0.09) differences were observed. Yet when females were separated out and examined in a mixed-effects model (with age, genotype and retention time as factors), a significant retention probe time x genotype difference was observed ($F_{(1,29)}$ = 13.54, p = 0.002). Posthoc analysis showed that both 2.5 and 7 month old female Tg26 +/- mice had significant short-term spatial memory losses at the 1 hour retention probe testing point, compared to same-aged female wild type -/- mice (Fig 5B). Males showed no significant short or long-term memory losses using this assay (Fig 5C). Therefore, only female Tg26 +/- mice showed short-term novel object location memory losses at both young adult and mature ages tested.

## Tg26 +/- female mice show mild learning deficits during acquisition trials in quadrant analysis of cued Barnes maze data

Barnes maze testing began at 3 months of age, with both spatial and olfactory cues used for Barnes Maze testing, as described in the methods. We first analyzed target quadrant results during the acquisition phase (i.e., days 1–4 of acquisition training; Fig 6). Young adult mice (3 months of age) showed latency to first head entrance (into the target quadrant) differences during acquisition training (Fig 6A) that was acquisition day dependent ($F_{(3,84)}$ = 2.72, p = 0.049) with female Tg 26 +/- mice showing, in general, higher latencies on acquisition days 1 and 2, relative to the other groups. No significant differences were seen in number of head entries into the target quadrant (Fig 6B). However, a significant time x sex interaction was observed for head time in the target quadrant ($F_{(3,81)}$ = 2.96, p = 0.04), with significantly less head time spent in the target quadrant by young adult female Tg 26 +/- mice on day 4 of acquisition, compared to female wild type mice (p<0.01; Fig 6C).

By 8 months of age during acquisition training (Fig 6D–6F), latency to first head entrance into the target quadrant differed by acquisition trial day ($F_{(3,81)}$ = 5.31, p = 0.002) and genotype ($F_{(1,27)}$ = 4.55, p = 0.04), with the now mature female Tg 26 +/- mice showing longer latencies in the target quadrant on days 2 and 3 of acquisition, compared to female wild type mice (p<0.01 each; Fig 6D). No other genotype or sex differences were seen during acquisition training at 8 month of age (Fig 6E and 6F). Thus, Tg26 (+/-) female mice showed mild acquisition deficits in target quadrant analysis of spatial and olfactory cued Barnes maze results at both young adult and mature ages.

## Tg26 +/- mice show spatial memory deficits during retention trials in target quadrant analyses of Barnes maze data

We next analyzed target quadrant results during retention probes at 24 hours and 7 days after the last acquisition trial (i.e., on days 5 and 12 of Barnes Maze testing, respectively; also Fig 6); for this, the escape tunnel was removed and replaced with a shallow tray similar to the other holes and the olfactory cues were removed. At 3 months of age (Fig 6A–6C), latency to first head entrance into the target quadrant showed a genotype difference ($F_{(1,29)}$ = 8.10, p = 0.0008). Post hoc assays indicate that young adult female Tg26 +/- mice had longer latencies at the 24 hour probe, compared to same-aged female wild type -/- littermates, while young adult male Tg26 +/- mice had longer latencies on probe day 7, compared to same-aged male wild type -/- littermates (p<0.01 each; Fig 6A), indicative of sex-specific short and long-term memory deficits in females and males, respectively. The number of head entries into the target quadrant showed probe time ($F_{(1,26)}$ = 16.88, p = 0.0004) and genotype differences ($F_{(1,28)}$ = 4.92, p = 0.03), with post hoc testing showing that young adult female Tg26 +/- mice had fewer

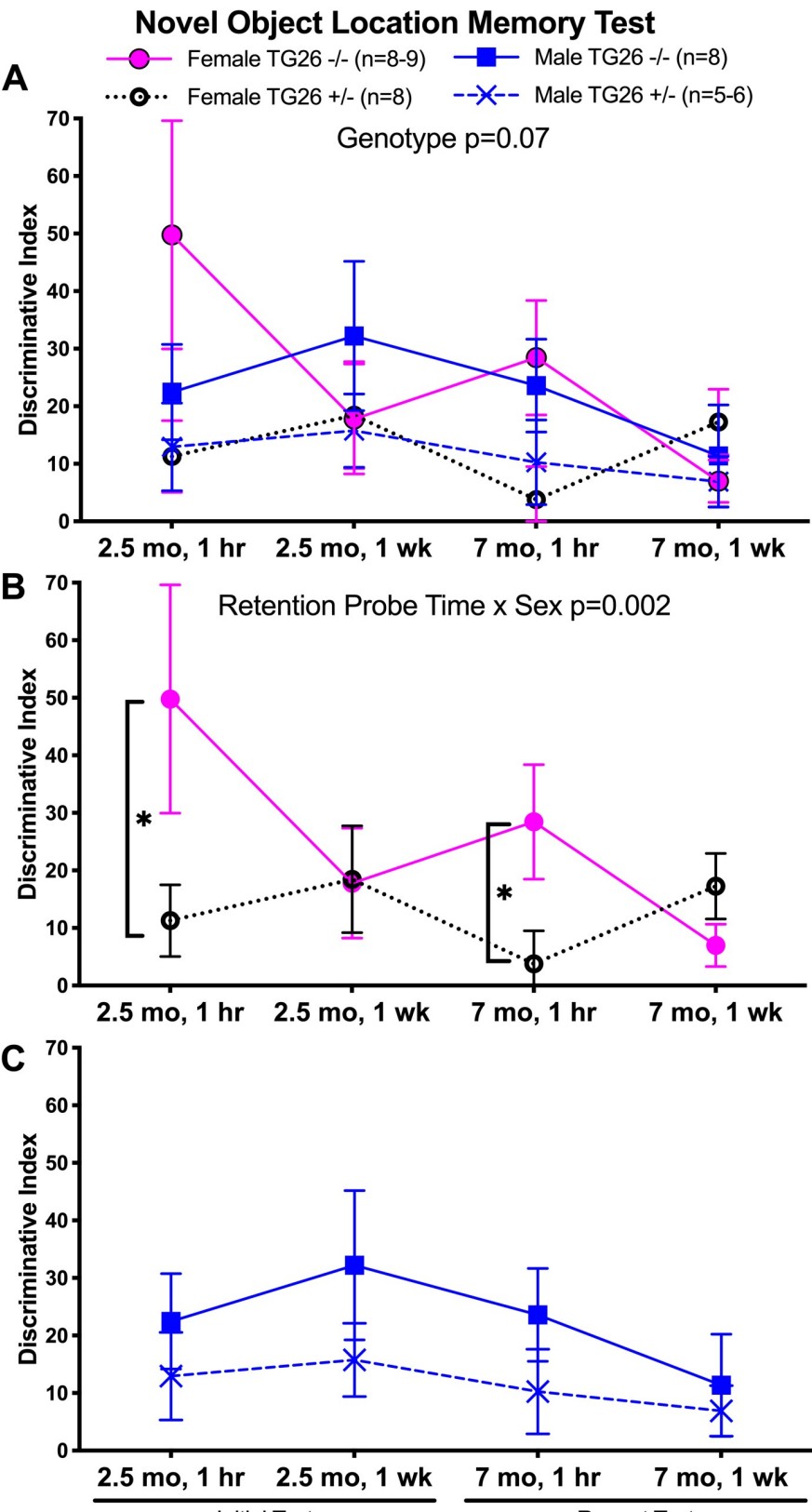

**Fig 5. Novel object location memory testing.** Assessment of spatial memory abilities in novel object location (NOL) assays in wild-type mice -/- and heterozygous Tg26 +/- mice of both sexes, at 2.5 and 7 months of age at two retention probe points (1 hour and 1 week after initial acquisition trial). (A-C) Results when tested in both sexes combined, females only, and males only, respectively. Mean ± SEM shown. n.s. = not significant. *: p<0.05, compared to wild type -/- mice of same sex. Numbers/group: At 3 months of age, there were 9 female wild type -/- mice, 8 female Tg26 +/- mice, 8 male wild type -/- and 7 male Tg26 +/- mice; At 8 months of age, there were 8 female wild type -/- mice, 8 female Tg26 +/- mice, 8 male wild type -/- and 6 male Tg26 +/- mice.

head entries at the 7 day probe time (day 12 of Barnes maze testing), compared same-aged female wild type -/- littermates (p<0.01; Fig 6B). Head time in the target quadrant showed retention probe time differences only ($F_{(1,27)}$ = 11.14, p = 0.002), but no post hoc test

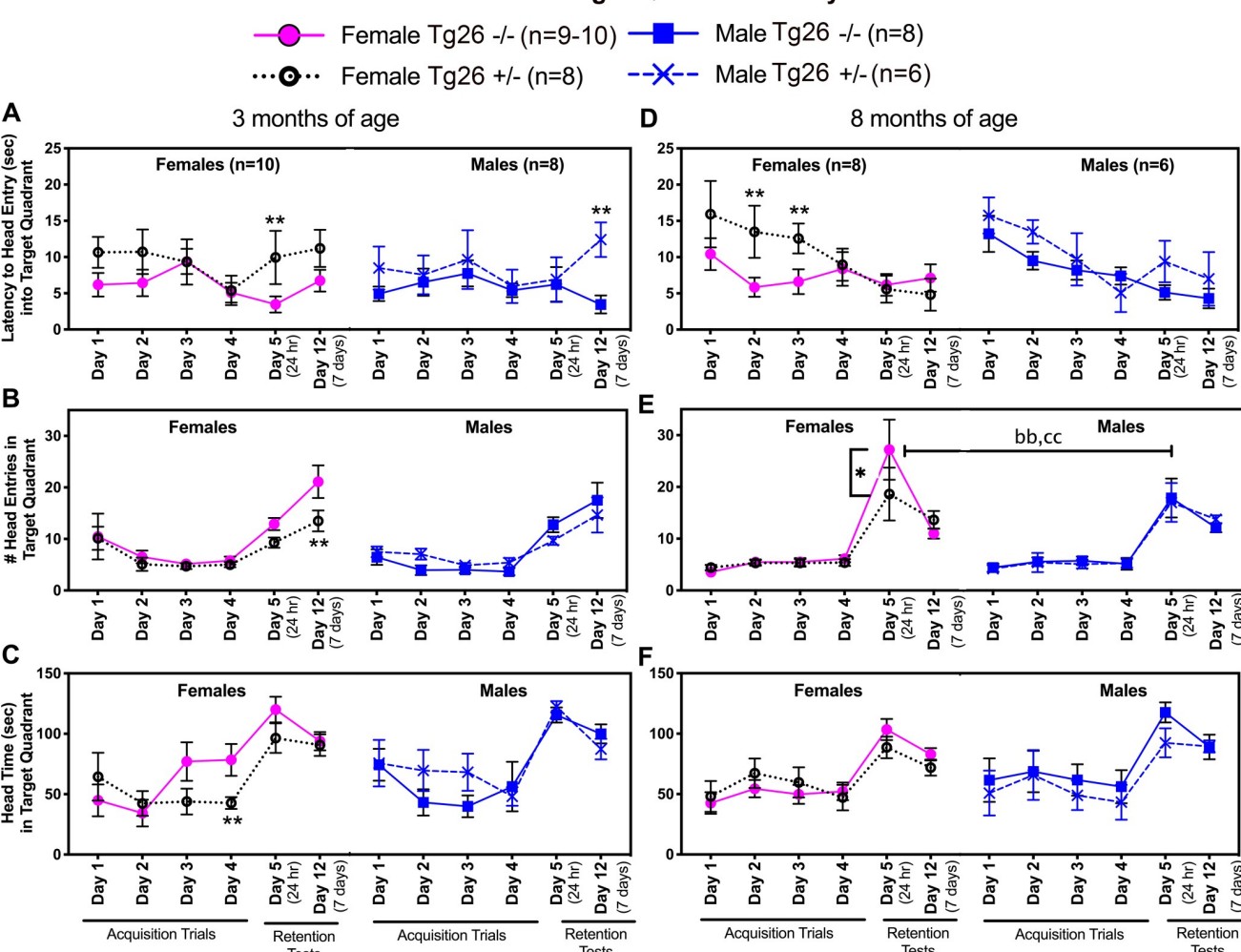

**Fig 6. Barnes maze target quadrant results.** Assessment of memory abilities on a Barnes maze at 3 and 8 months of age, during target quadrant analyses of four acquisition days (Days 1–4 of Barnes maze testing) and at two retention probe test points (24 hours and 7 days after completion of acquisition training, which are Days 5 and 12 of Barnes maze testing). (A and D) Latency to first head entry into the target quadrant at 3 and 8 months of age, respectively. (B and E) Number (#) of head entries into target quadrant at 3 and 8 months of age, respectively. (C and F) Head time in target quadrant at 3 and 8 months of age, respectively. Mean ± SEM shown. *:p<0.05 and **: p<0.01, compared to wild type -/- mice of same sex. bb: p<0.01, compared to opposite sex of same genotype (opposite sex). cc: p<0.01, compared to opposite sex of opposite genotype (opposite sex and genotype). For ease of reading, mix-model effects findings are indicated in the text. Numbers/group: At 3 months of age, there were 10 female wild type -/- mice, 8 female Tg26 +/- mice, 8 male wild type -/- and 6 male Tg26 +/- mice. At 8 months of age, there were 9–10 female wild type -/- mice, 8 female Tg26 +/- mice, 8 male wild type -/- and 6 male Tg26 +/- mice.

differences (Fig 6C). Hence, young adult female Tg26 +/- mice at this early time point showed mild indices of both short and long-term memory losses, while the young adult male Tg26 +/- mice showed only one mild long-term deficit (latency), during target quadrant analyses.

At 8 months of age (Fig 6D–6F), no sex or genotype differences were observed for latency to first head entrance into the target quadrant (Fig 6D) or head time in the target quadrant (Fig 6F). However, there were differences in the number of head entries that was retention probe time ($F_{(1,52)}$ = 7.55, p = 0.008) and genotype dependent ($F_{(1,52)}$ = 4.05, p = 0.049). Post hoc comparisons indicated that the now mature female Tg26 +/- mice had fewer head entries into the target quadrant at the 24 hour probe test, compared to mature female wild type mice (p<0.01; Fig 6E). Interestingly, female wild type mice had more head entries than male mice of either genotype (p<0.01 each, Fig 6E). Therefore, mature female Tg26 +/- mice at the 8-month time point showed mild short-term memory losses during target quadrant analyses, compared to mature female wild type mice.

## Tg26 (+/-) female mice have short and long-term spatial memory deficits in target hole analyses of Barnes maze data

In order to explore the memory deficits further, we next analyzed target hole at 24 hours and 7 days after the final acquisition trial with both genotypes, sexes and time (age at time of testing and memory retention probe times) combined into mixed-effect models. The main effect for number of head entries into the target hole (Fig 7A) was time ($F_{(2,61)}$ = 5.92, p = 0.002) and genotype ($F_{(1,29)}$ = 4.95, p = 0.03). Post hoc analysis indicated that young adult female Tg26 +/- mice had the lowest number of head entries at the 3 month, 24 hour probe point, compared to same-aged female and male wild-type mice (p<0.05 each). The remaining assays (head time, duration, latency, distance, and path efficiency to the target hole) showed differences only in mature (8 month old mice), as described next.

Head time in the target hole (Fig 7B) also showed time ($F_{(2,81)}$ = 4.98, p = 0.01) and genotype ($F_{(1,107)}$ = 4.81, p = 0.03) differences, with mature female Tg26 +/- mice having the lowest head time (p<0.05) at the 8 month, 24 hour probe, compared to same-aged male wild type mice. Also, mature male wild type mice showed higher head time at the 8 month, 24 hr probe than at their 3 month, 24 hour, and 8 month, 7 day probe timepoints (Fig 7B). Duration of visits to the target hole (Fig 7C) showed main effects of time ($F_{(3,74)}$ = 4.15, p = 0.01) and genotype ($F_{(1,30)}$ = 4.48, p = 0.04), with both mature female and male Tg26 +/- mice showing shorter visits to the target hole at the 8 month, 24 hr probe point, compared to same-aged male wild type mice (p<0.0.5 each). The mature male wild type mice also showed longer durations than at their 3 month, 24 hr and 8 month, 7 day probe points (Fig 7C). Latency to first head entrance into the target hole showed no main effects in the mixed model (Fig 7D). Yet, when the 8 month, 7 day time point data was assayed using a two-way ANOVA with genotype and age as factors, female Tg26 +/- mice showed higher latencies than the other groups (p<0.01 or p<0.05, as shown in Fig 7D). The main effects for distance to first head entry (Fig 7E) were time ($F_{(3,78)}$ = 3.12, p = 0.04) and sex ($F_{(1,29)}$ = 4.81, p = 0.03), with post hoc assays showing mature female Tg26 +/- mice traveled longer distances at the 8 month, 7 day probe point, compared to same-aged male Tg26 +/- mice (p<0.05).

Lastly, examination of the path efficiency to first entry into the target hole (Fig 8A; assayed by the AnyMaze software) showed differences by time ($F_{(2,78)}$ = 3.48, p = 0.03) and sex ($F_{(1,108)}$ = 8.93, p = 0.003). A post hoc analysis showed that mature female Tg26 +/- mice had the lower path efficiency at the 8 month, 7 day probe point, compared to the other groups (p<0.05 each). A manual scoring of the strategy used by the mice to reach the target hole showed that a higher proportion of the mature female Tg26 +/- mice at 8 months, 7 day probe point, chose a

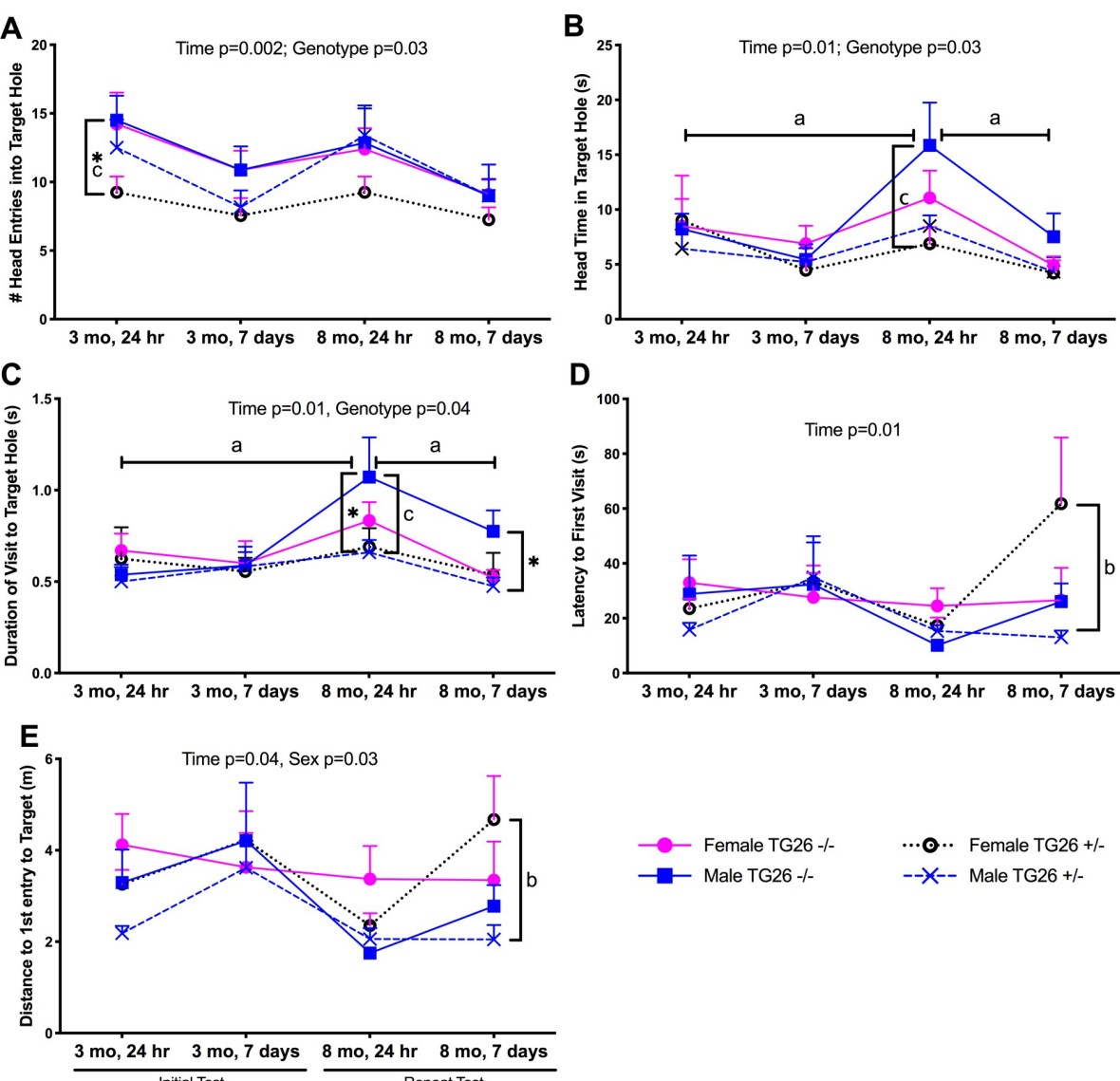

**Fig 7. Barnes maze target hole results.** Assessment of memory abilities on a Barnes maze during target hole analyses at 3 and 8 months of age, at two retention probe test points (24 hours and 7 days after completion of acquisition training, which are Days 5 and 12 of Barnes maze testing). (A) Mean number of head entries into the target hole. (B) Mean head time at target hole in seconds (s). (C) Mean duration of each visit to target hole. (D) Mean latency of first visit to target hole. (E) Mean distance to first entrance of head into target hole in meters (m). Mean ± SEM shown. *: $p<0.05$, compared to wild type -/- mice of same sex; a<0.05, compared to values obtained for these same animals at the other time points indicated; b: $p<0.05$, compared to opposite sex of same genotype (opposite sex); c: $p<0.05$, compared to opposite sex of opposite genotype (opposite sex and genotype). Numbers/group as indicated in Fig 6.

random strategy rather than the more typically serial or direct pathways used by the other groups (Fig 8B). Fig 8C shows representative images of path efficiency to the target hole by animals from each group. While male Tg26 -/- mice often showed direct routes to the target hole, female Tg26 +/- mice often showed a long serial path in the wrong direction first, delaying their first arrival at the target hole.

In summary, Figs 7 and 8 reveal that female Tg26 +/- mice showed the greatest number of Barnes Maze target hole spatial memory deficits, with only a mild short-term deficit as young

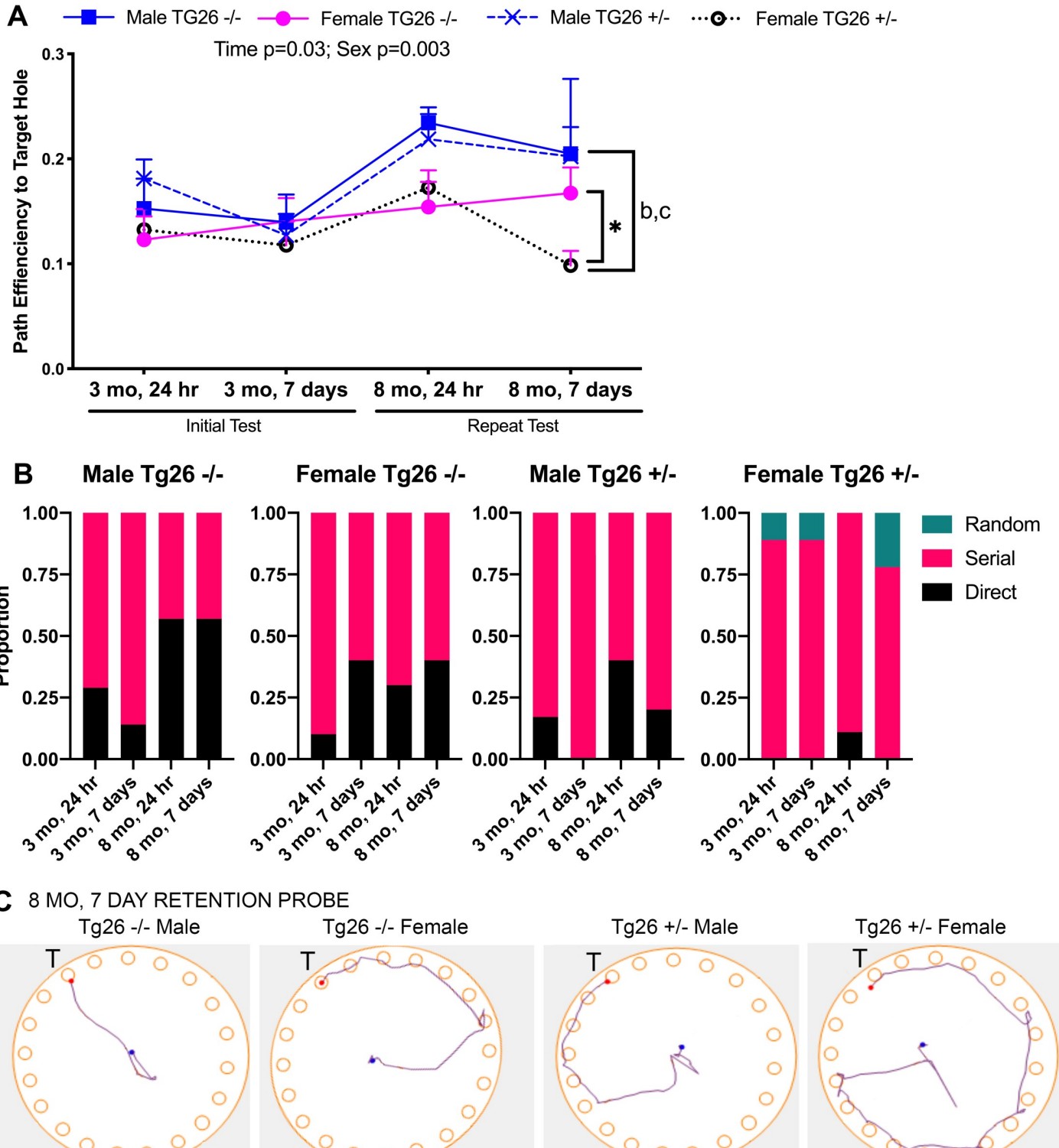

**Fig 8. Barnes maze target hole path efficiency.** Assessment of memory abilities on a Barnes maze during target hole analyses at 3 and 8 months of age, at two retention probe test points. (A) Mean path efficiency to the target hole, in which the lower number is a lower efficiency, as assayed by the AnyMaze software. Mean ± SEM shown. *: p<0.05, compared to wild type -/- mice of same sex; a<0.05, compared to values obtained for these same animals at the other time points indicated; b: p<0.05, compared to opposite sex of same genotype (opposite sex); c: p<0.05, compared to opposite sex of opposite genotype (opposite sex and genotype). (B) The strategies used by the mice at each testing time was manually scored as direct, serial or random, and is presented as a proportion per group. (C) Figures showing representative path efficiency plots from the 8 month, 7 day, retention probe. T = Target hole. Numbers/group as indicated in Fig 6.

adults, yet a number of both short and long term spatial memory deficits at maturity (i.e., at 8 months of age).

To aid interpretation of the results, total distance traveled and speed on the Barnes maze was also measured (Fig 9). When both genotypes, sexes, and time were combined into a mixed-effect model, the total distance traveled around the Barnes Maze showed time ($F_{(4,105)}$ = 62.49, p<0.0001) and genotype x time interaction differences ($F_{(4,105)}$ = 3.42, p = 0.01; Fig 9A). Each genotype and sex showed increased distances traveled at each subsequence time point examined, compared to their 3 month, Day 1 results (p<0.01 each), but no genotype differences in the post hoc assays. When male mice were separated out and examined in a mixed-effects model (with age, genotype and retention time as factors), the time difference was still observed ($F_{(4,43)}$ = 26.83, p<0.0001; Fig 9B); similar results were seen when female mice were separated out and examined ($F_{(4,62)}$ = 36.34, p<0.0001; Fig 9C). Post hoc assays of each group showed an increase in distance traveled at each subsequence time point examined, compared to their 3 month, Day 1 results (p<0.01 each, showed in Fig 9B and 9C).

With regard to speed around the maze, when both genotypes, sexes, and time were combined into a mixed-effect model, speed around the Barnes Maze during testing showed time ($F_{(4,103)}$ = 3.36, p = 0.02) and genotype x time interaction differences ($F_{(4,103)}$ = 3.30, p = 0.02; Fig 9D). Posthoc analysis showed overall that female Tg26 +/- and male Tg26 +/- mice were slower than male wild type mice (p<0.02 and p = 0.04, respectively). When male mice were separated out and examined in a mixed-effects model, a significant genotype difference was observed ($F_{(1,14)}$ = 3.32, p = 0.49; Fig 9E), with male Tg26 +/- mice traveling slower on the maze than male wild type -/- mice on Acquisition Day 0 (Fig 9E), suggestive of a higher level of uncertainty in the male Tg26 +/- mice when deciding to enter the target hole. When female mice were separated out and examined in a mixed-effects model, no significant differences in speed were observed (Fig 9F).

## Both TG26 +/- sexes show mild to moderate spatial memory heading errors

To assess whether Tg26(+/-) mice recognized the target hole in relation to the other holes on the maze, the number of times the mice entered each hole was measured during the short term (24 hours) and long term (14 days) probe sessions. This reference error metric is a robust and reliable method for assessment of hippocampal-dependent spatial memory [44, 45]. This same method was also used in a recent paper from the lab using a different cohort of Tg26 -/- and +/- mice [3], thus, allowing for comparison to that prior study. For this assay, heading errors into all holes of the Barnes maze were assessed at 3 and 8 months of age (Fig 10). At the 3 month, 24 hour retention probe time point, with both genotypes, sexes and hole combined in the mixed-effects model, the main effect was hole ($F_{(19,304)}$ = 18.69, p<0.0001; Fig 10A). Post hoc assays revealed overall genotype differences in the number of head entries into the target hole (Tg26 -/- > +/-, p<0.05), and that young adult female Tg26 +/- mice showed fewer head entries into the target hole and more visits to holes on the opposite side of the maze (Left (L) 9 and Right (R) 9), compared to same-aged female wild type mice (p<0.05 each, Fig 10B), indicative of short-term spatial memory deficits that were more prevalent in young adult female Tg26 +/- mice.

At the 3 month, 7 day retention probe (day 12 of testing; Fig 10B), now hole x genotype interaction differences were observed ($F_{(19,53)}$ = 1.74, p = 0.02; the hole difference was $F_{(9, 245)}$ = 8.03, p<0.0001). Post hoc assays again showed overall genotype differences in number of head entries into the target hole (Tg26 -/- > +/-, p<0.05), with young adult female Tg26 +/- mice having fewer head entries into the target hole (p<0.01) and more head entries into the opposite L9 hole (p<0.01). The young adult male Tg26 +/- mice showed fewer entrances into

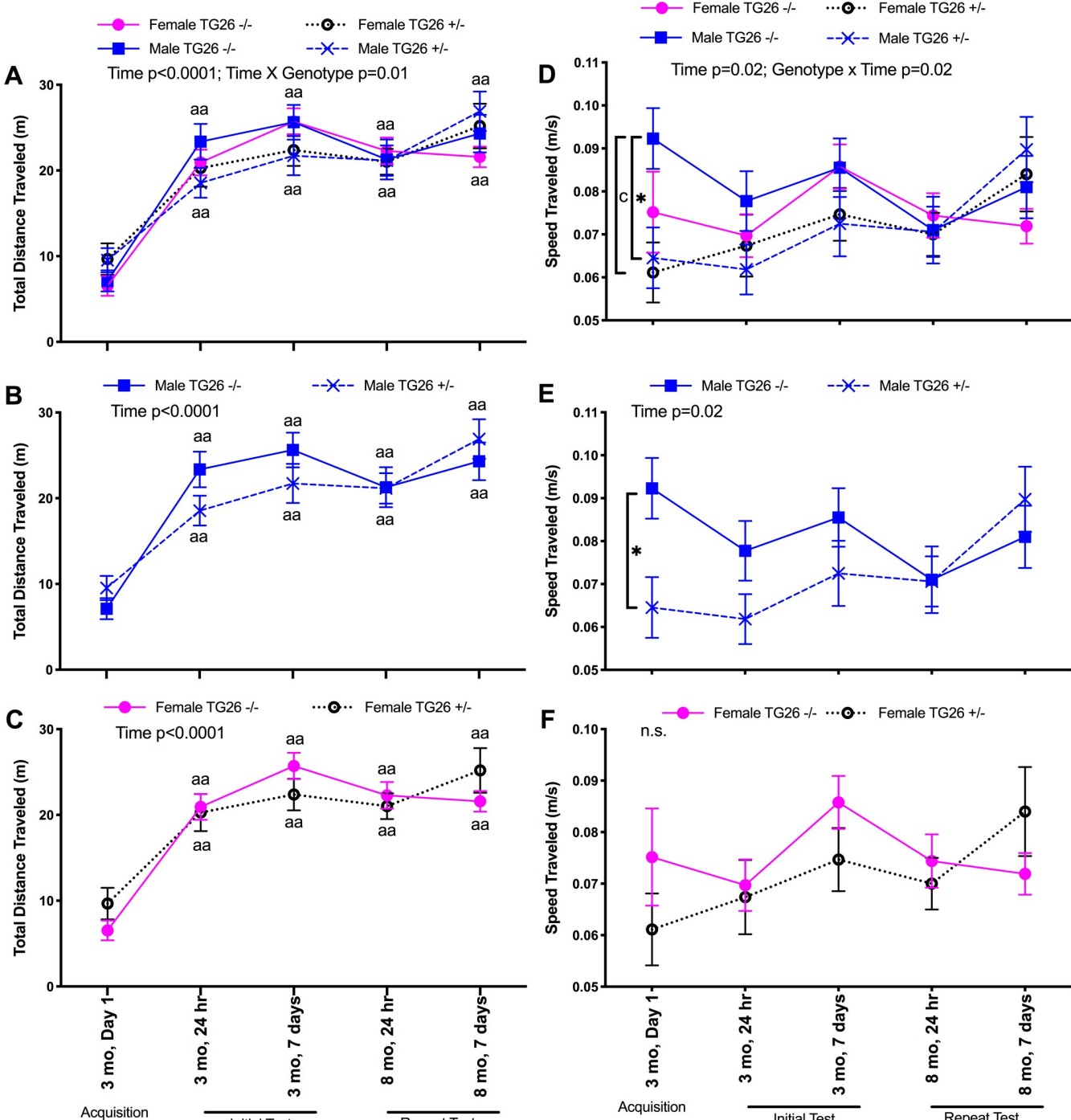

**Fig 9. Total distance traveling and speed on the Barnes maze.** Assessment of total distance and mean speed on the Barnes maze at 3 and 8 months of age, at two retention probe test points (24 hours and 7 days after completion of acquisition training, which are Days 5 and 12 of Barnes maze testing). (A) Mean total distance on the apparatus during a 300 second Barnes maze test in meters (m). (B) Mean speed on the apparatus in meters per second (m/s). *: p<0.05, compared to wild type -/- mice of same sex; aa: p<0.05 and p<0.01, compared to values obtained for these same animals at 3 months, Day 1; c: p<0.05, compared to opposite sex of opposite genotype. Numbers/group as indicated in Fig 6.

the R3 hole, compared to same-aged male wild type mice (p<0.01, Fig 10B). These latter changes are suggestive of long-term spatial memory deficits in both young adult female and male Tg26 +/- mice.

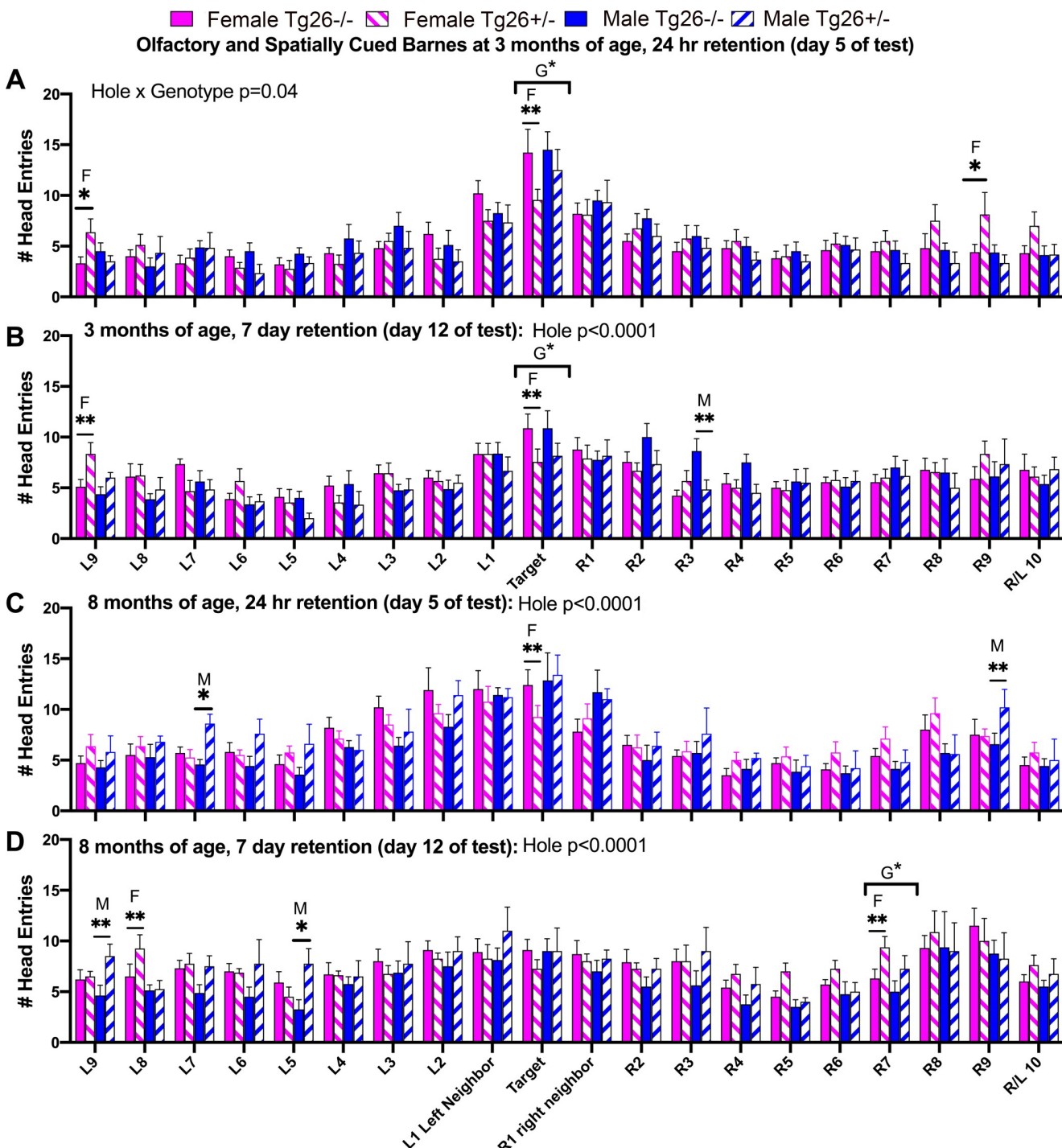

**Fig 10. Barnes maze heading error results for all holes.** Number (#) of head entries into each hole of the Barnes maze at 3 or 8 months of age, on retention days at the 24 hour or 7 day retention probe time points (days 5 and 12 after completion of acquisition training, respectively). (A) Results at 3 month, 24 hour retention probe point. (B) Results at 3 month, 7 day retention probe point. (C) Results at 8 month, 24 hour retention probe point. (D) Results at 8 month, 7 day retention probe point. Mean ± SEM shown. *: p<0.05 and **: p<0.01, compared to wild type -/- mice of same sex, with males differences indicated with an M, and female differences indicated with a F. G*: p<0.05 genotype difference. Numbers/group as indicated in Fig 6.

By 8 months of age, at the 24 hour retention probe, hole ($F_{(19,494)}$ = 15.19, p<0.0001) differences were again observed (Fig 10C). Post hoc assays revealed that mature female Tg26 +/- mice visited the target hole less than age-matched female wild type mice (p<0.01), while mature male Tg26 +/- mice visited the opposite L7 and R9 holes more than same-aged male wild type mice (p<0.05 and p<0.01, respectively). At the 8 month, 7th day retention probe, again hole ($F_{(19,494)}$ = 5.40, p<0.0001) differences were observed (Fig 10D). Post hoc assays showed a general genotype difference for the R7 hole (Tg26 +/- > -/-, p<0.05), that mature female Tg26 +/- mice visited the R7 and L8 holes more than aged-matched female wild type mice (p<0.01 each) and that mature male Tg26 +/- mice visited the L9 and L5 holes more than mature male wild type mice (p<0.01 and p<0.05, each; Fig 10D). These 8-month results are indicative of both short and long-term spatial memory deficits in the now mature female Tg26 +/- mice, and several heading errors into non-target holes by mature male Tg26 +/- mice.

## Only biological sex-based differences were seen in tibial bone attributes

Degenerative changes in bone have been observed in mature HIV-1 Tg heterozygous rats (NL4-3Δgag/pol Fisher344 rats), compared to mature wild type rats [46, 47]. Thus, on study completion, tibial bones were collected at 11 months of age and the proximal ends assayed using ex vivo microCT to determine if there were bone degenerative changes in mature HIV Tg mice as a result of the general physical characteristics or the observed neurological issues. Only sex-related based differences were observed in percent trabecular bone volume (BV/TV; Fig 11A) and trabecular number (Tb.N.; Fig 11D) were observed, with males showing higher percent bone volume and higher trabecular number than females of both genotypes. No differences were observed in trabecular thickness, trabecular separation or trabecular tissue mineral density), or in cortical bone mineral density (Fig 11B, 11C, 11E and 11F). Post hoc analyses showed differences only in trabecular number, with males of each genotype showing higher numbers of trabeculae than females of each genotype (Fig 11D). Both percent trabecular bone volume and trabecular number correlated positively and significantly with body weight (BV/TV changes correlated with body weight: r = 0.38, p = 0.03; Tb. N changes correlated with body weight: r = 0.43, p = 0.02), suggesting that the greater weights in males was a key factor to the higher bone volumes and trabecular numbers in male mice, regardless of genotype.

## Discussion

These data demonstrate both peripheral neuronal and central neurocognitive declines in HIV-1 Tg26 +/- mice, using for the first time a longitudinal design for both sexes and for all assays except for the tibial bone attributes. Sex-related differences were seen, with female HIV-1 Tg26 +/- mice displaying greater peripheral neuronal dysfunction and earlier neurocognitive memory declines than male Tg26 +/- counterparts. Female HIV-1 Tg26 +/- mice displayed progressive hot temperature hyposensitivity (hyposensitivity to 41°C at 2.5 months of age that progressed to also include hyposensitivity to 45°C by 10 months of age), compared to wild type littermates. The female HIV-1 Tg26 +/- mice showed consistent moderate short and long-term spatial memory deficits, as well as spatial learning deficits, at both 3 and 8 months of age, although these deficits were more apparent at 8 months of age when examining data from the more specific Barnes maze quadrant and hole analyses, than the novel object location memory assay. In contrast, the male HIV-1 Tg26 +/- mice showed only mild long-term spatial memory deficits at 3 months of age, and heading errors into non target holes by 8 months of age. Even with the latter increase, 8 month old male Tg26 +/- mice showed fewer memory deficits than 8 month old female Tg26 +/- mice. In summary, greater sex-specific temperature

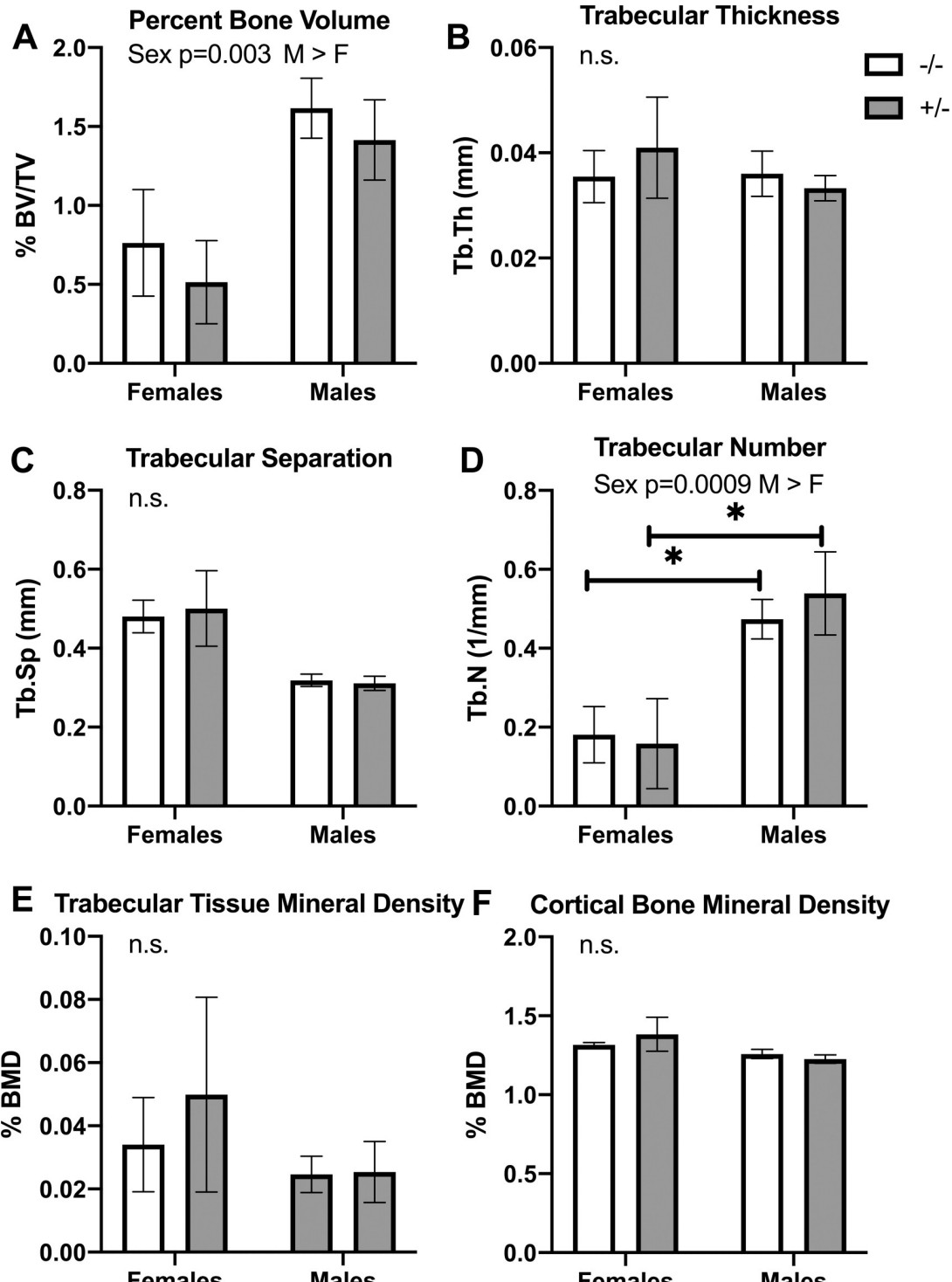

**Fig 11. Micro-CT analysis of proximal tibia of Tg26 +/- mice and their wild type (-/-) littermates after collection at 11 months of age.** (A-E) Histograms show the trabecular parameters of tibia: percent bone volume/tissue volume (% BV/TV), trabecular thickness (Tb.Th), trabecular separation (Tb.Sp), Trabecular number (Tb.N) and trabecular bone mineral density (BMD). (F) Histogram showing cortical bone mineral density in the proximal tibia. Mean ± SEM shown. *: p<0.05, compared to wild type -/- mice of same sex. Numbers/group were: 6 female wild type -/- mice, 5 female Tg26 +/- mice, 8 male wild type -/- and 6 male Tg26 +/- mice.

hyposensitivity and spatial memory declines were observed in female HIV Tg26 +/- mice, than in male Tg26 +/- mice, or their wild type littermates, that increased with age.

Sex-specific differences in the control of cognitive behaviors have been documented in the literature [48, 49]. While the influence of the estrous cycle was not addressed in this current study, since cognitive behaviors were tested multiple times, in multiple litters, using more than one method over a period of 300 days (10 months), it is unlikely that estrous cycle is the sole driver of our cognitive findings. That said, sex hormones are known to contribute to neuropathic pain behaviors in C57BL/6 male and female mice that received multiple intrathecal injections of HIV-glycoprotein 120 (gp120) [50]. The gp120 injections in female mice lead to enhanced mechanical and cold sensitivity, compared to their male counterparts. Such enhanced mechanical and cold sensitivity were reduced in ovariectomized gp120-injected female mice (heat sensitivity was not examined in the Guindon et al, 2019 study). Although, the influence of sex hormones on noxious heat hyposensitivity is not known, it may be an additional contributing factor in the sex-specific thermal hyposensitivity findings observed in the female Tg+/- mice since tactile hyposensitivity has been reported in felines infected with the feline immunodeficiency virus (FIV) [51]. Distal sensory polyneuropathy is a frequent finding with lentivirus infections (e.g., HIV and FIV) of the peripheral nerve system, and is one of the most common neurological disorders in patients in HIV infections [52]. The signs and symptoms in these patients include tactile hyposensitivity and other sensory losses. Although the underlying mechanism is still unclear, it may be due to neuronal damage in dorsal root ganglia after activation and upregulation of proteinase-activated receptor 1 by interleukin 1beta [51].

Similar to the HIV-1 Tg rats, we observed no significant general health disparities relative to wild type animals [5, 15, 17, 53]. After the initial slow speeds by male Tg26 +/- mice during the acquisition testing period, Tg26 +/- mice of either sex did not display speed differences in the Barnes maze apparatus and good motor competence, matching results from HIV-1 Tg rats [10, 11]. Thus, there were no observed illnesses that could confound the assessment of learning and cognition in the Tg26 +/- mice.

Sex-specific neurocognitive differences were observed in a cross-sectional study of mature (8–10 month old) Tg26+/- mice from our research core [3], as well as in HIV Tg mice in Morris water maze testing [54]. The hippocampi of the mature female Tg26 +/- mice in the prior Putatunda study showed greater reductions in numbers of neural progenitor cells, compared to mature male Tg26 +/- mice (although both groups had fewer quiescent neural stem cells and neuroblasts than age-matched wild type mice) [3]. These same mature female Tg26 +/- mice had impaired learning and short-term spatial memory in Barnes maze testing, while the mature male Tg26 +/- mice showed short- and long-term spatial memory deficits, compared to wild type mice. In this current study, we were able to extend the literature on HIV Tg26 mice considerably by using a longitudinal design (examining mice from 1–10 months of age), and observed progressive spatial memory deficits in female HIV Tg26 +/- mice, particularly in their ability to find the target hole efficiently during Barnes maze testing. We must note that we may have missed the chance to detect memory deficits in the novel object location test by performing the retention probe test at 1 week after the acquisition test (the time was chosen to match the Barnes maze testing points). Back to the point, in HIV-1 Tg rats, sex-specific temporal processing differences are more pronounced in HIV-1 female rats than in HIV-1 male rats [6, 25]. Female HIV-1 rats have also been shown to have significant neurocognitive deficits that appear early in development [6, 55]. As in HIV-1 Tg rats [9, 17], Tg26 +/- mice, particularly the females, showed mild to moderate cognitive deficits across the functional lifespan indicative of the permanence of the effect of HIV-1 proteins. Also similar to HIV-1 Tg rats that underwent Morris water maze testing [10], we observed longer latencies in trial

acquisition learning phases (in female Tg26 +/- only mice here, however), retention phase testing, and inefficient path efficiency to target quadrants. Sex differences have also been identified in humans with HAND [56, 57], which combined with our findings in Tg26 mice and results from HIV-1 rat models, add to the growing amount of literature showing sex-dimorphic differences in cognition and HIV infection.

However, in contrast to findings in studies showing lower weight gains in HIV Tg rats due to a lower buildup of lean body mass [15, 17, 58], we did not observe genotype differences in weight in the HIV Tg26 +/- mice, compared to same-sex and age-matched wild type mice. We also did not observe increased incidence of tumors in Tg26 +/- mice, in contrast to findings from HIV-1 Tg rats [25, 55]. Additionally, no clear indices of increased anxiety was observed in this longitudinal study or in the initial cross-sectional study of HIV Tg26 +/- mice [3], in contrast to HIV-1 Tg rats that show increased perimeter exploration over center field exploration, indicative of enhanced anxiety [17]. In this current study, the 10 month long period of handling and testing gave ample opportunity for the mice to habituate to the tester (R.L. in each case), the room and the testing apparati, giving an underlying reason for a lack of anxiety in these Tg26 +/- mice. This lack of anxiety is further supported by the touch escape results, in which the mice began crawling into the tester's hands rather than escaping during the 7 month testing point. Further studies of HIV Tg26 +/- mice are needed to determine if they have similar altered anxiety and exploratory behaviors, attention deficit disorders, auditory temporal processing deficits, or reverse maze learning deficits, as seen in HIV-1 Tg rats [9, 59].

Regarding tibial bone changes, we observed only sex-specific changes in tibias of mature male mice (increased bone volume and trabecular number), compared to mature female mice, regardless of their genotype. This increase correlated with the differential increases in body weight, regardless of genotype (males gained more weight). These results differ from studies examining bone microarchitecture or osteoclast function in mature HIV-1 Tg rats (NL4-3Δgag/pol Fisher344 rats), compared to mature wild types [46, 47]. HIV Tg rats showed severe bone destruction as a consequence of dysregulated osteoclastogenesis in hindlimb bones [46, 47]. HIV-1 Tg rats (NL4-3Δgag/pol Fisher344 rats) also showed low gains in body weight from a very early age and throughout life due to a reduced build-up of skeletal muscle [60]. A consistently lower body mass is associated with lower weight-induced mechanical loading of bone and therefore lower bone growth [61]. Lower skeletal muscle mass may also contribute to the lower bone mass, since skeletal muscle is a key secretory endocrine organ involved in the regulation of metabolism, including bone metabolism [62–64]. Lastly, the HIV-1 Tg rats used in the prior study (NL4-3Δgag/pol Fisher344 rats) developed organ failure and earlier death than wild type counterparts by their second year of life [15, 65]. Thus, our findings of no bone degeneration and a continued increase in body weight highlights that there are differences in the HIV Tg mice versus rats. Organ failure, such as chronic kidney disease, can lead to significant bone loss and quality [66]. We have followed HIV Tg26 +/- female and male mice for up to 21 months (unreported findings). Female HIV Tg26 +/- mice show consistently shiny coats and no fur loss across this time frame, and while male HIV Tg26 +/- mice become heavier due to more fat and more hair loss than male wild types, no organ failure has been noted. We suggest that future studies consider examining underlying reasons for these differences.

In conclusion, greater sex-specific temperature hyposensitivity and spatial memory declines were observed in female HIV Tg26 +/- mice, primarily due to faulty spatial strategies such as the random searches when mature, than in male Tg26 +/- mice, or their wild type littermates, that increased with age. Also, there were only a few differences in outcomes of HIV Tg26 +/- mice versus HIV Tg rats, at least for the behavioral outcomes and phenotypic characteristics examined in this study. These results combined with our other recent publication indicate that

HIV-1 Tg26 mice is a promising model in which to study neuropathic mechanisms underlying peripheral pathology as well as cognitive deficits seen with HIV.

## Supporting information

**S1 Fig. Transfer cage behaviors.** Remaining transfer cage behaviors from general health screen testing, in wild-type mice -/- and heterozygous Tg26 +/- mice of both sexes, across time (1, 2 and 7 months of age). No significant differences (n.s.) were observed between groups for these behaviors.
(TIF)

**S2 Fig. Neurological reflexive testing.** Remaining general neurological health screenings results that were not shown in Fig 2, in wild-type mice -/- and heterozygous Tg26 +/- mice of both sexes, across time (1, 2 and 7 months of age). No significant differences (n.s.) were observed between groups for these behaviors.
(TIF)

**S1 Data.**
(XLSX)

## Author Contributions

**Conceptualization:** Mary F. Barbe, Regina Loomis, Jennifer Gordon.

**Data curation:** Mary F. Barbe, Regina Loomis.

**Formal analysis:** Mary F. Barbe, Regina Loomis, Adam M. Lepkowsky, Huaqing Zhao.

**Funding acquisition:** Jennifer Gordon.

**Investigation:** Mary F. Barbe, Regina Loomis, Adam M. Lepkowsky, Steven Forman.

**Methodology:** Mary F. Barbe, Regina Loomis.

**Project administration:** Mary F. Barbe.

**Supervision:** Mary F. Barbe, Jennifer Gordon.

**Writing – original draft:** Mary F. Barbe, Regina Loomis.

**Writing – review & editing:** Mary F. Barbe, Regina Loomis, Jennifer Gordon.

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
