## [Decision Letter · Decision Letter 0]

23 Apr 2020

PONE-D-20-04655

A longitudinal characterization of sex-specific somatosensory and spatial memory deficits in HIV Tg26 heterozygous mice

PLOS ONE

Dear Dr. Barbe,

Thank you for submitting your manuscript to PLOS ONE. After careful consideration by a Reviewer and an Academic Editor, all of the critiques must be addressed in detail in a revision to determine publication status. If you are prepared to undertake the work required, I would be pleased to reconsider my decision, but revision of the original submission without directly addressing the critiques of the Reviewer does not guarantee acceptance for publication in PLOS ONE. If the authors do not feel that the queries can be addressed, please consider submitting to another publication medium. A revised submission will be sent out for re-review. The authors are urged to have the manuscript given a hard copyedit for syntax and grammar.

Comments to the Author

1. Is the manuscript technically sound, and do the data support the conclusions?

Reviewer #1: Yes

2. Has the statistical analysis been performed appropriately and rigorously? 

Reviewer #1: Yes

3. Have the authors made all data underlying the findings in their manuscript fully available?

Reviewer #1: Yes

4. Is the manuscript presented in an intelligible fashion and written in standard English?

Reviewer #1: Yes

5. Review Comments to the Author

Reviewer #1: In this paper, the authors carried out a longitudinal study to characterize sensory and spatial memory deficits in a non-infectious model of HIV-1 (Tg26 mice). This HIV transgenic line contains an integrated transgene encoding the entire NL4-3 HIV-1 genome with a deletion of the gag/pol genes (on a C57Bl/6J background). The stability of the transgene on this background allowed the authors to test its effects across the lifespan of the animals, thus assessing the progression of HIV associated neurological disorders (HAND) with age. This is a significant advancement of previous studies that have used a cross sectional design to study HAND in either young or aged mice. In addition, the authors studied the progression of HAND in males versus female mice. They found that while both sexes did not show abnormal reflexive responses or weight loss due to viral gene expression, both males and female transgenic mice developed hyposensitivity to heat at 10 months of age. However, this hyposensitivity is evident in females even at 2.5 months of age. The authors then used 2 behavioral tasks to assess spatial memory – novel object location and the Barnes maze. The authors again saw an effect of gender, with female transgenic mice showing more severe deficits in spatial memory compared to males and manifesting at the earlier age of 3 months of age compared to 8 months in males. Finally, the authors did not find any change in tibial bone attributes due to viral gene expression. In conclusion, this study highlights important differences in the progression of HIV induced neurological deficits in male versus female mice as well as establishes the HIV-1 Tg26 mouse as a promising model of HAND.

Major points:

1. Given that the authors use a number of subjective scoring techniques (the SHIRPA behaviors), they need to give a more detailed description of what constitutes an ‘abnormal’ response in each situation and how these were translated to a quantitative score (especially in Fig. 2F and Supplementary Fig.1).

2. The authors need to describe how the discriminative index score was calculated for the novel object location test. The authors should also indicate the chance level of novel object location exploration. With the possible exception of 2.5 mo Tg26-/- male mice, all the other groups have poor discriminative scores at the 1 week time point. Without a probabilistic measure of chance exploration, we cannot determine if these mice remember the novel location at all. Also, given that most studies using this paradigm use 24h as a measure of long term memory, why did the authors test 7 days later?

3. While the authors have used a variety of parameters to study the spatial memory of mice in the Barnes maze, they have not included some of the more standard measures, such as total errors or strategy use during the retention tests. Given the genotype differences in running speed and distance travelled seen in a previous study of these mice (Putatunda et al. 2019, Ref. 3) as well as the variability in running speed observed in Supplementary fig 3. , the above parameters may be a more accurate measure of spatial memory than latency to the target hole. Measurements of strategy use (direct search versus serial search) would especially add value to the conclusion made in page 22 line 22.

4. The rationale behind analyzing tibial bones in the context of this study is unclear. The HIV Tg26 mouse line has been reported to show a range of symptoms including lymphomas, cardiomyopathy and inflammation as the authors have noted in their introduction. The authors’ decision to examine osteoclastogenesis as part of a study that focuses on the cognitive deficits associated with HIV needs to be further explained.

Minor points:

1. The authors bred the Tg26 line by back crossing the original strain from the FVB/N background to a C57Bl/6J background. Did they analyze the genome of the new strain for SNP variations to confirm the C57Bl/6J background?

2. Page 7, Line 10-13 of the Methods section states that the Irwin Observation Test battery was used to measure indices of neuropsychiatric function. Page 7, Line 2 (legend for Fig. 1) then introduces the term SHIRPA, which is a behavioral screen developed in 1997 based on the Irwin battery. If the authors are using the SHIRPA screen (i.e. modified from the original Irwin battery) for their study, they should state this clearly in their methods, name the test in full and cite the original paper.

3. Page 8, Line 20. The authors should elaborate on what constitutes ‘spontaneous behaviors’ and how these were quantitatively scored.

4. Page 9, Line 16-17. The authors need to elaborate on how the ‘magnitude of reaction’ was quantified and translated into the discriminative indices shown in Fig. 6.

5. The results in Fig. 5, the authors have described differences in the discriminative index of female Tg26-/- mice. Were there any differences in the total time these mice interacted with all the objects? This could be another measure of anxiety as well as confound the interpretation of their discriminative ability.

6. Fig. 5C – The legend shows male Tg26 -/- mice are indicated with blue squares. However the figure uses blue circles to represent this group.

7. Fig. 6E – The figure legend does not describe the * shown in this figure. What comparison does this signify?

8. Page 21, Line 18-19. “Results are organized by time of detection of significant differences.” This statement is unclear.

9. Fig. 7 – The authors need to elaborate on how path efficiency was calculated.

10. Page 24-25, Lines 21-5. At 8 months of age, wild type and transgenic mice of both sexes seem unable to distinguish the target hole versus any other hole in the Barnes maze (Fig. 8D). The non-target holes should not have any spatial valence for the mice and hence entry into these should be entirely a matter of chance. Given the above, I disagree with the authors that differences in entries into non-target holes at this time point (8 mo, 1 week) is indicative of any spatial memory deficits. The differences seen by the authors could be due to unforeseen underlying preferences or an artefact of a small sample size. As mentioned before, a calculation of errors would be a better indicator of spatial memory than the heading errors used in this study.

6. PLOS authors have the option to publish the peer review history of their article (what does this mean?). If published, this will include your full peer review and any attached files.

Do you want your identity to be public for this peer review? For information about this choice, including consent withdrawal, please see our Privacy Policy.

Reviewer #1: No

We would appreciate receiving your revised manuscript by October, 2020. To enhance the reproducibility of your results, we recommend that if applicable you deposit your laboratory protocols in protocols.io, where a protocol can be assigned its own identifier (DOI) such that it can be cited independently in the future. For instructions see: http://journals.plos.org/plosone/s/submission-guidelines#loc-laboratory-protocols

We look forward to receiving your revised manuscript.

Kind regards,

Stephen D. Ginsberg, Ph.D.

Section Editor

PLOS ONE

---

## [Author Response · Author response to Decision Letter 0]

1 Dec 2020

November 24, 2020

Re: PONE-D-20-04655

“A longitudinal characterization of sex-specific somatosensory and spatial memory deficits in HIV Tg26 heterozygous mice”

Editor:

Responses to Reviewer #1

Reviewer #1: In this paper, the authors carried out a longitudinal study to characterize sensory and spatial memory deficits in a non-infectious model of HIV-1 (Tg26 mice). This HIV transgenic line contains an integrated transgene encoding the entire NL4-3 HIV-1 genome with a deletion of the gag/pol genes (on a C57Bl/6J background). The stability of the transgene on this background allowed the authors to test its effects across the lifespan of the animals, thus assessing the progression of HIV associated neurological disorders (HAND) with age. This is a significant advancement of previous studies that have used a cross sectional design to study HAND in either young or aged mice. In addition, the authors studied the progression of HAND in males versus female mice. They found that while both sexes did not show abnormal reflexive responses or weight loss due to viral gene expression, both males and female transgenic mice developed hyposensitivity to heat at 10 months of age. However, this hyposensitivity is evident in females even at 2.5 months of age. The authors then used 2 behavioral tasks to assess spatial memory – novel object location and the Barnes maze. The authors again saw an effect of gender, with female transgenic mice showing more severe deficits in spatial memory compared to males and manifesting at the earlier age of 3 months of age compared to 8 months in males. Finally, the authors did not find any change in tibial bone attributes due to viral gene expression. In conclusion, this study highlights important differences in the progression of HIV induced neurological deficits in male versus female mice as well as establishes the HIV-1 Tg26 mouse as a promising model of HAND.

Response: Thank you for the nice synopsis. 

Major points:

1. Given that the authors use a number of subjective scoring techniques (the SHIRPA behaviors), they need to give a more detailed description of what constitutes an ‘abnormal’ response in each situation and how these were translated to a quantitative score (especially in Fig. 2F and Supplementary Fig.1).

Response: A detailed explanation of the scoring method was provided in a newly provided Table 1 and more explanation into the text of the methods.

2. The authors need to describe how the discriminative index score was calculated for the novel object location test. The authors should also indicate the chance level of novel object location exploration. With the possible exception of 2.5 mo Tg26-/- male mice, all the other groups have poor discriminative scores at the 1 week time point. Without a probabilistic measure of chance exploration, we cannot determine if these mice remember the novel location at all. Also, given that most studies using this paradigm use 24h as a measure of long term memory, why did the authors test 7 days later?

Response: 

A description of the discriminative index score has been added to the methods section on page 10: “Discrimination between the objects was calculated using a discrimination ratio, calculated as the absolute difference in the time spent exploring the novel and familiar objects, divided by the total time spent exploring the objects [1-3]. This calculation takes into account the individual differences in the total amount of exploration [1-3].” This method reduces the possibility of chance exploration. 

Timing of novel object location tests: We used a time choice of one week to match the Barnes timing. We may have missed a best long-term retention time by the wild type mice. The reason for the timing was added to the methods, and the potential weakness of this design was added to the discussion.

3. While the authors have used a variety of parameters to study the spatial memory of mice in the Barnes maze, they have not included some of the more standard measures, such as total errors or strategy use during the retention tests. Given the genotype differences in running speed and distance travelled seen in a previous study of these mice (Putatunda et al. 2019, Ref. 3) as well as the variability in running speed observed in Supplementary fig 3. , the above parameters may be a more accurate measure of spatial memory than latency to the target hole. Measurements of strategy use (direct search versus serial search) would especially add value to the conclusion made in page 22 line 22.

Response: 

a. We appreciate the suggestion. Total errors have now been calculated and are shown below. Unfortunately, as shown by several other researchers [4-8], a calculation of total errors into the target hole was less sensitive than a measurement based on locating the escape hole using a nose to head defection (a primary error measurement) at the first encounter of the mouse with the escape hole. Also, as suggested by O’Leary in 2011 [8], mice trained with cues make fewer errors overall than mice trained without cues. Since our mice were all trained on the Barnes maze with olfactory cues, the total error measurements may be altered by the provision of cues and does not appear to accurately reflect what the mouse has learned. We added the cues part to our discussion. However, we did not add the less sensitive [in this study at least] “total error” calculation to the manuscript.

THIS FIGURE IS AVAILABLE IN THE RESPONSE ATTACHED WITH THE INDIVIDUAL FILES.

b. Distance around the Barnes maze per testing time point, and the velocity used around the Barnes maze, findings were moved from the Supplemental data section and is now presented in a new Figure 9. This data was reassessed from the raw data and then re-analyzed more stringently using the same 3 step statistical strategu as the other Barnes Maze data (all together first in a 3 way repeated measures model, and then after separating out by sex). This improved the findings and now shows that male Tg26 -/- mice moved more slowly around the maze during the acquisition Day 0 phase, similar to Putatunda et al, 2019 findings during the during the acquisition (see below). 

c. To clarify the reviewer’s comments on the Putatunda et al 2019 findings versus the results of this study, in the Putatunda study, significant differences were observed only during the acquisition phases. In that study, we observed that female Tg26 +/- mice had longer distances but similar speeds during the initial acquisition phase, as female wild type mice; in contrast, male Tg26 +/- mice traveled the same distance, but at a slower speed, than the male wild type mice. This matches our Figure 9D and E results exactly. Further, no genotype differences were observed between the sexes during the retention/probe phases in Putatunda et al 2019, again matching our current findings exactly. 

d. Measurements of strategy use (direct search versus serial search) would especially add value to the conclusion made in page 22 line 22. 

Response: 

We did present the strategy used during the retention tests in the first submission. It was presented as “path efficiency”. It is a quantitative AnyMaze software measurement of the direct versus serial search. The original “path efficiency” as measured by AnyMaze is the same method that we used in Putatunda et al, 2019 for reporting the search strategy and is necessary to keep for continuity between the studies. In this revision, we kept our original “path efficiency” as measured by AnyMaze in what is now Figure 8A , and page 24. We also added a manual method of strategy search and results (see below).

We have now defined path efficiency” in the methods page 12 and below in response # 9. It is defined as: “Path efficiency is a measure performed by the AnyMaze software and is an index of the efficiency of the path taken by the animal to get from the first position in the test (i.e., the center site of the maze where they are placed initially) to the last position (i.e., the target). For this, the software divides the straight line distance between the first position in the test and the last position by the total distance travelled by the animal during the test. A value of 1 indicates perfect efficiency - the animal moved in a straight line indicative of a “direct search” - values less than 1 indicate decreasing efficiency. The lowest value would be indicative of the most inefficient “serial search”.

Yet, to meet the reviewer’s concerns, the proportion of animals per group exhibiting a direct, serial versus random strategy for reaching the target hole was also manually scored from the AnyMaze plot tracts. These new results are shown in Figure 8B, and described on pages 12 and 24. 

e. Also, to meet the reviewer’s concerns, we also added a summary of this finding to the conclusion.

4. The rationale behind analyzing tibial bones in the context of this study is unclear. The HIV Tg26 mouse line has been reported to show a range of symptoms including lymphomas, cardiomyopathy and inflammation as the authors have noted in their introduction. The authors’ decision to examine osteoclastogenesis as part of a study that focuses on the cognitive deficits associated with HIV needs to be further explained.

Response: We did not study osteoclastogenesis (no histology was performed). Instead, we novelly forwarded the characterization of bone morphology in Tg26 mice (similar to kidney, cardiac and bone morphology in past studies of Tg26 rats). 

The rationale was expanded in the Introduction page 5, “As a consequence, mice heterozygous for the transgene develop many clinical features of HIV infection and AIDS including a disease similar to HIV-associated nephropathy (HIVAN), as well as changes in other tissues and organs, including, inflammation, B cell lymphomas, skin lesions consistent with cutaneous papillomas, cardiomyopathy, and muscle wasting between 3 and 6 months of age on the FVB/N background [9-12]. Interestingly, degenerative changes in bone, another key organ, have been observed in mature HIV-1 Tg heterozygous rats (NL4-3�gag/pol Fisher344 rats), compared to mature wildtype rats [13, 14]. However, possible bone degeneration that may occur as a result of altered general physical characteristics or neurological issues has yet to be examined in HIV mouse models.”

We also added rationale to the results section on page 29: “Degenerative changes in bone have been observed in mature HIV-1 Tg heterozygous rats (NL4-3�gag/pol Fisher344 rats), compared to mature wildtype rats [13, 14]. Thus, on study completion, tibial bones were collected at 11 months of age and the proximal ends assayed using ex vivo microCT to determine if there were bone degenerative changes in mature HIV Tg mice as a result of the general physical characteristics or the observed neurological issues.”

We would also like to direct the reviewer to page 34 of the Discussion where more is discussed on bone and other organ, and body weight differences between HIV Tg mice versus rats. We expanded this paragraph some as well. 

Minor points:

1. The authors bred the Tg26 line by back crossing the original strain from the FVB/N background to a C57Bl/6J background. Did they analyze the genome of the new strain for SNP variations to confirm the C57Bl/6J background?

Response: The reviewer has a good suggestion for the future. Unfortunately, we did not examine the back-crossed animals for SNP variations. But we did perform traditional backcrosses for a minimum of 6 generations in order to establish the C57Bl/6J background, as mentioned in the text on page 6, line 9 -11. We also cited our past paper using the same back crossing method [15].

2. Page 7, Line 10-13 of the Methods section states that the Irwin Observation Test battery was used to measure indices of neuropsychiatric function. Page 7, Line 2 (legend for Fig. 1) then introduces the term SHIRPA, which is a behavioral screen developed in 1997 based on the Irwin battery. If the authors are using the SHIRPA screen (i.e. modified from the original Irwin battery) for their study, they should state this clearly in their methods, name the test in full and cite the original paper.

Response: We have re-written this section of the methods to be more clear (General health and neurological testing) and added a new Table 1 showing the exact tests used and their scoring methods. We are unsure which author(s) specifically the reviewer wants us to cite, as several authors modified the original Irwin screening method [16] in parallel in 1997 [17, 18]. Also, that 1997 modified SHIRPA version has been further modified and expanded by many researchers across the years in order to enhance the effectiveness of these screenings. Therefore, we also cited those papers of expansion and/or explanation as well (also needed since of the few of the earlier papers were not as clear regarding the methods as some of the latter) [19-25].

We provided a new Table 1 indicated exactly which tests were performed and the scoring system used. We also expanded in the methods on the subtest that we had to modify (the touch escape) in order to address the habituation of the mice to either the tester or the test across time.

3. Page 8, Line 20. The authors should elaborate on what constitutes ‘spontaneous behaviors’ and how these were quantitatively scored.

Response: A very detailed explanation of the scoring method was provided in a newly provided Table 1, and the text of the methods.

4. Page 9, Line 16-17. The authors need to elaborate on how the ‘magnitude of reaction’ was quantified and translated into the discriminative indices shown in Fig. 6.

Response: A discriminative index score was used. Its description has been added to the methods section, as were references: “Discrimination between the objects was calculated using a discrimination ratio, calculated as the absolute different in the time spent exploring the novel and familiar objects, divided by the total time spent exploring the objects [1-3]. This calculation takes into account the individual differences in the total amount of exploration [1-3].”

5. The results in Fig. 5, the authors have described differences in the discriminative index of female Tg26-/- mice. Were there any differences in the total time these mice interacted with all the objects? This could be another measure of anxiety as well as confound the interpretation of their discriminative ability.

Response: There was no significant difference in total time the mice interacted with all of the objects, between the groups in this longitudinal study. The touch escape results also indicate that there was no anxiety. The longitudinal nature of this study likely reduced anxiety in the mice as they were exposed to the same tester and testing room from 1 months of age for another 10 months.

We added this to the discussion: “In this current study, the 10 month long period of handling and testing gave ample opportunity for the mice to habituate to the tester (R.L. in each case), the room and the testing apparati, giving an underlying reason for a lack of anxiety in these Tg26 +/- mice. This lack of anxiety is further supported by the touch escape results, in which the mice began crawling into the tester’s hands rather than escaping during the 7 month testing point.”

6. Fig. 5C – The legend shows male Tg26 -/- mice are indicated with blue squares. However, the figure uses blue circles to represent this group.

Response: Thank you. Fixed. 

7. Fig. 6E – The figure legend does not describe the * shown in this figure. What comparison does this signify?

Response: Thank you. This has been added to that figure legend: “*:p<0.05 and **: p<0.01, compared to wild type -/- mice of same sex.” 

8. Page 21, Line 18-19. “Results are organized by time of detection of significant differences.” This statement is unclear.

Response: We removed this statement altogether as it was not really needed. 

9. Fig. 7 – The authors need to elaborate on how path efficiency was calculated.

Response: The following was added to the methods: “Path efficiency is a measure performed by the AnyMaze software and is an index of the efficiency of the path taken by the animal to get from the first position in the test (i.e., the center site of the maze where they are placed initially) to the last position (i.e., the target). For this, the software divides the straight line distance between the first position in the test and the last position by the total distance travelled by the animal during the test. A value of 1 indicates perfect efficiency - the animal moved in a straight line indicative of a “direct search” - values less than 1 indicate decreasing efficiency. The lowest value would be indicative of the most inefficient “serial search”.

10. Page 24-25, Lines 21-5. At 8 months of age, wild type and transgenic mice of both sexes seem unable to distinguish the target hole versus any other hole in the Barnes maze (Fig. 8D). The non-target holes should not have any spatial valence for the mice and hence entry into these should be entirely a matter of chance. Given the above, I disagree with the authors that differences in entries into non-target holes at this time point (8 mo, 1 week) is indicative of any spatial memory deficits. The differences seen by the authors could be due to unforeseen underlying preferences or an artefact of a small sample size. As mentioned before, a calculation of errors would be a better indicator of spatial memory than the heading errors used in this study.

a. Response: We appreciate the suggestion. Total errors have now been calculated and are shown above. Unfortunately, as shown by several other researchers [4-8], a calculation of total errors into the target hole was less sensitive than a measurement based on locating the escape hole using a nose to head defection (a primary error measurement) at the first encounter of the mouse with the escape hole. Also, as suggested by O’Leary in 2011 [8], mice trained with cues make fewer errors overall than mice trained without cues. Since our mice were all trained on the Barnes maze with olfactory cues, the total error measurements may be altered by the provision of cues and does not appear to accurately reflect what the mouse has learned. Thus, we did not add the less sensitive [in this study at least] “total error” calculation to the manuscript.

We also added the same explanation for the test used, as we had placed in the Putatunda et al 2019 paper. We apologize for not including it in the first submission of this paper: “To assess whether Tg26(+/-) mice recognized the target hole in relation to the other holes on the maze, the number of times the mice entered each hole was measured during the short term (24 hours) and long term (14 days) probe sessions. This reference error metric is a robust and reliable method for assessment of hippocampal-dependent spatial memory [4, 26]. This same method was also used in a recent paper from the lab using a different cohort of Tg26 -/- and +/- mice [15], thus, allowing for comparison to that prior study.”

References:

1. Dix SL, Aggleton JP. Extending the spontaneous preference test of recognition: evidence of object-location and object-context recognition. Behav Brain Res. 1999;99(2):191-200. Epub 1999/10/08. doi: 10.1016/s0166-4328(98)00079-5. PubMed PMID: 10512585.

2. Ennaceur A, Delacour J. A new one-trial test for neurobiological studies of memory in rats. 1: Behavioral data. Behav Brain Res. 1988;31(1):47-59. Epub 1988/11/01. doi: 10.1016/0166-4328(88)90157-x. PubMed PMID: 3228475.

3. Vogel-Ciernia A, Wood MA. Examining object location and object recognition memory in mice. Curr Protoc Neurosci. 2014;69:8 31 1-17. Epub 2014/10/10. doi: 10.1002/0471142301.ns0831s69. PubMed PMID: 25297693; PubMed Central PMCID: PMCPMC4219523.

4. Harrison FE, Reiserer RS, Tomarken AJ, McDonald MP. Spatial and nonspatial escape strategies in the Barnes maze. Learn Mem. 2006;13(6):809-19. doi: 10.1101/lm.334306. PubMed PMID: 17101874; PubMed Central PMCID: PMCPMC1783636.

5. Garcia MF, Gordon MN, Hutton M, Lewis J, McGowan E, Dickey CA, et al. The retinal degeneration (rd) gene seriously impairs spatial cognitive performance in normal and Alzheimer's transgenic mice. Neuroreport. 2004;15(1):73-7. Epub 2004/04/27. doi: 10.1097/00001756-200401190-00015. PubMed PMID: 15106834.

6. O'Leary TP, Brown RE. The effects of apparatus design and test procedure on learning and memory performance of C57BL/6J mice on the Barnes maze. J Neurosci Methods. 2012;203(2):315-24. Epub 2011/10/11. doi: 10.1016/j.jneumeth.2011.09.027. PubMed PMID: 21982740.

7. O'Leary TP, Brown RE. Optimization of apparatus design and behavioral measures for the assessment of visuo-spatial learning and memory of mice on the Barnes maze. Learn Mem. 2013;20(2):85-96. Epub 2013/01/17. doi: 10.1101/lm.028076.112. PubMed PMID: 23322557.

8. O'Leary TP, Savoie V, Brown RE. Learning, memory and search strategies of inbred mouse strains with different visual abilities in the Barnes maze. Behav Brain Res. 2011;216(2):531-42. Epub 2010/08/31. doi: 10.1016/j.bbr.2010.08.030. PubMed PMID: 20801160.

9. Curreli S, Krishnan S, Reitz M, Lunardi-Iskandar Y, Lafferty MK, Garzino-Demo A, et al. B cell lymphoma in HIV transgenic mice. Retrovirology. 2013;10:92. Epub 2013/08/30. doi: 10.1186/1742-4690-10-92. PubMed PMID: 23985023; PubMed Central PMCID: PMCPMC3847158.

10. Serrano AL, Jardi M, Suelves M, Klotman PE, Munoz-Canoves P. HIV-1 transgenic expression in mice induces selective atrophy of fast-glycolytic skeletal muscle fibers. Front Biosci. 2008;13:2797-805. Epub 2007/11/06. doi: 10.2741/2886. PubMed PMID: 17981754.

11. Villarroya J, Diaz-Delfin J, Hyink D, Domingo P, Giralt M, Klotman PE, et al. HIV type-1 transgene expression in mice alters adipose tissue and adipokine levels: towards a rodent model of HIV type-1 lipodystrophy. Antivir Ther. 2010;15(7):1021-8. Epub 2010/11/03. doi: 10.3851/IMP1669. PubMed PMID: 21041917.

12. Lewis DK, Callaghan M, Phiri K, Chipwete J, Kublin JG, Borgstein E, et al. Prevalence and indicators of HIV and AIDS among adults admitted to medical and surgical wards in Blantyre, Malawi. Trans R Soc Trop Med Hyg. 2003;97(1):91-6. Epub 2003/07/31. doi: 10.1016/s0035-9203(03)90035-6. PubMed PMID: 12886812.

13. Vikulina T, Fan X, Yamaguchi M, Roser-Page S, Zayzafoon M, Guidot DM, et al. Alterations in the immuno-skeletal interface drive bone destruction in HIV-1 transgenic rats. Proc Natl Acad Sci U S A. 2010;107(31):13848-53. Epub 2010/07/21. doi: 10.1073/pnas.1003020107. PubMed PMID: 20643942; PubMed Central PMCID: PMCPMC2922243.

14. Lafferty MK, Fantry L, Bryant J, Jones O, Hammoud D, Weitzmann MN, et al. Elevated suppressor of cytokine signaling-1 (SOCS-1): a mechanism for dysregulated osteoclastogenesis in HIV transgenic rats. Pathog Dis. 2014;71(1):81-9. Epub 2014/01/01. doi: 10.1111/2049-632X.12117. PubMed PMID: 24376119; PubMed Central PMCID: PMCPMC4048640.

15. Putatunda R, Zhang Y, Li F, Fagan PR, Zhao H, Ramirez SH, et al. Sex-specific neurogenic deficits and neurocognitive disorders in middle-aged HIV-1 Tg26 transgenic mice. Brain Behav Immun. 2019;80:488-99. Epub 2019/04/19. doi: 10.1016/j.bbi.2019.04.029. PubMed PMID: 30999016; PubMed Central PMCID: PMCPMC6660421.

16. Irwin S. Comprehensive observational assessment: Ia. A systematic, quantitative procedure for assessing the behavioral and physiologic state of the mouse. Psychopharmacologia. 1968;13(3):222-57. Epub 1968/09/20. PubMed PMID: 5679627.

17. Crawley JN, Paylor R. A proposed test battery and constellations of specific behavioral paradigms to investigate the behavioral phenotypes of transgenic and knockout mice. Hormones and behavior. 1997;31(3):197-211. Epub 1997/06/01. doi: 10.1006/hbeh.1997.1382. PubMed PMID: 9213134.

18. Rogers DC, Fisher EM, Brown SD, Peters J, Hunter AJ, Martin JE. Behavioral and functional analysis of mouse phenotype: SHIRPA, a proposed protocol for comprehensive phenotype assessment. Mamm Genome. 1997;8(10):711-3. PubMed PMID: 9321461.

19. Hatcher JP, Jones DN, Rogers DC, Hatcher PD, Reavill C, Hagan JJ, et al. Development of SHIRPA to characterise the phenotype of gene-targeted mice. Behav Brain Res. 2001;125(1-2):43-7. Epub 2001/10/30. doi: 10.1016/s0166-4328(01)00275-3. PubMed PMID: 11682092.

20. Crawley JN. Behavioral phenotyping of transgenic and knockout mice: experimental design and evaluation of general health, sensory functions, motor abilities, and specific behavioral tests. Brain Res. 1999;835(1):18-26. Epub 1999/08/17. doi: 10.1016/s0006-8993(98)01258-x. PubMed PMID: 10448192.

21. Crawley JN. What's wrong with my mouse? : behavioral phenotyping of transgenic and knockout mice. 2nd ed. Hoboken, N.J.: Wiley-Interscience; 2007. xvi, 523 p. p.

22. Crawley JN. Behavioral phenotyping strategies for mutant mice. Neuron. 2008;57(6):809-18. Epub 2008/03/28. doi: 10.1016/j.neuron.2008.03.001. PubMed PMID: 18367082.

23. Masuya H, Inoue M, Wada Y, Shimizu A, Nagano J, Kawai A, et al. Implementation of the modified-SHIRPA protocol for screening of dominant phenotypes in a large-scale ENU mutagenesis program. Mamm Genome. 2005;16(11):829-37. Epub 2005/11/15. doi: 10.1007/s00335-005-2430-8. PubMed PMID: 16284798.

24. Paylor R, Spencer CM, Yuva-Paylor LA, Pieke-Dahl S. The use of behavioral test batteries, II: effect of test interval. Physiology & behavior. 2006;87(1):95-102. Epub 2005/10/04. doi: 10.1016/j.physbeh.2005.09.002. PubMed PMID: 16197969.

25. Jacquelin C, Strazielle C, Lalonde R. Neurologic function during developmental and adult stages in Dab1(scm) (scrambler) mutant mice. Behav Brain Res. 2012;226(1):265-73. Epub 2011/09/29. doi: 10.1016/j.bbr.2011.09.020. PubMed PMID: 21945093.

26. Attar A, Liu T, Chan WT, Hayes J, Nejad M, Lei K, et al. A shortened Barnes maze protocol reveals memory deficits at 4-months of age in the triple-transgenic mouse model of Alzheimer's disease. PLoS One. 2013;8(11):e80355. doi: 10.1371/journal.pone.0080355. PubMed PMID: 24236177; PubMed Central PMCID: PMCPMC3827415.

---

## [Decision Letter · Decision Letter 1]

16 Dec 2020

A longitudinal characterization of sex-specific somatosensory and spatial memory deficits in HIV Tg26 heterozygous mice

PONE-D-20-04655R1

Dear Dr. Barbe,

We’re pleased to inform you that your manuscript has been judged scientifically suitable for publication and will be formally accepted for publication once it meets all outstanding technical requirements.

Kind regards,

Stephen D. Ginsberg, Ph.D.

Section Editor

PLOS ONE

**Comments to the Author**

1. If the authors have adequately addressed your comments raised in a previous round of review and you feel that this manuscript is now acceptable for publication, you may indicate that here to bypass the “Comments to the Author” section, enter your conflict of interest statement in the “Confidential to Editor” section, and submit your "Accept" recommendation.

Reviewer #1: All comments have been addressed

2. Is the manuscript technically sound, and do the data support the conclusions?

Reviewer #1: Yes

3. Has the statistical analysis been performed appropriately and rigorously? 

Reviewer #1: Yes

4. Have the authors made all data underlying the findings in their manuscript fully available?

Reviewer #1: Yes

5. Is the manuscript presented in an intelligible fashion and written in standard English?

Reviewer #1: Yes

6. Review Comments to the Author

Reviewer #1: The authors have addressed all points of concern and made significant edits to their text and figures. In particular, the author's expansion of the methods used to score behavioral tests and addition of key controls for the same has improved the flow and substance of the paper. I recommend this article for publication in PLOS One.

7. PLOS authors have the option to publish the peer review history of their article (what does this mean?). If published, this will include your full peer review and any attached files.

Reviewer #1: No

---

## [Editor Report · Acceptance letter]

22 Dec 2020

PONE-D-20-04655R1 

A longitudinal characterization of sex-specific somatosensory and spatial memory deficits in HIV Tg26 heterozygous mice 

Dear Dr. Barbe:

I'm pleased to inform you that your manuscript has been deemed suitable for publication in PLOS ONE. Congratulations! Your manuscript is now with our production department. 

Kind regards, 

on behalf of

Dr. Stephen D. Ginsberg 

Section Editor

PLOS ONE